

# Enhanced understanding of dominant drivers of Water Yield change across China through the improved coupled carbon and water model

Huilan Shen [1, 2], Hanbo Yang [1, 2, *], Changming Li [1,2,3]

[1] Department of Hydraulic Engineering, Tsinghua University, Beijing 100084, China

[2] State Key Laboratory of Hydroscience and Engineering, Tsinghua University, Beijing 100084, China

[3] School of Civil Engineering and Transportation, State Key Laboratory of Subtropical Building and Urban Science, South China University of Technology, Guangzhou 510641, China

* Correspondence: Hanbo Yang (yanghanbo@tsinghua.edu.cn)





**Abstract:** The rapid environmental changes, including climate change, escalating
atmospheric $CO_2$ concentration ($[CO_2]$), and vegetation dynamics, have been
significantly impacting hydrological processes. Accurately quantifying their
contribution to water yield (WY) has become a significant challenge in water resource
management and climate adaptation studies. Therefore, this study improved the coupled
carbon and water (CCW) model integrating dynamic water use efficiency (WUE) to
quantify the $CO_2$-physiological feedback; furthermore systematically investigated the
causes for WY change during 1982-2017 in China using a scenario analysis method
based on the improved CCW model. The results showed that the effects on WY from
changes in climate, vegetation, and $[CO_2]$ exhibited a significant regional variability.
Climate change (especially precipitation change) emerged as the dominant driver,
directly affecting over 70% of China's land area. The vegetation change was the second
largest factor, especially in central China, where vegetation change led to a general
decrease in runoff. The effect of the escalating $[CO_2]$, which reduced transpiration by
inducing stomatal closure, was relatively small. Spatial analysis aligned with isohyetal
lines further revealed that vegetation change and $[CO_2]$ exerted greater influence within
the 400–1600 mm precipitation range. In addition, the elasticity analysis showed that
the sensitivity ranking of impact factors is precipitation ($\varepsilon P = 1.55$) > $[CO_2]$ ($\varepsilon CO_2 =$
0.55) > NDVI ($\varepsilon NDVI = -0.44$) for the whole China. Historically, NDVI change has
exceeded precipitation and $[CO_2]$ impacts on runoff in some regions due to its higher
relative change; however, CMIP6 SSP585 projections indicate that accelerating $[CO_2]$
rise (2.34% $yr^{-1}$) will amplify its hydrological effect to a +1.29% annual WY increase
by 2100, surpassing vegetation influences. This study provided theoretical support for
water resource management and offered new perspectives for climate change
adaptation strategies, vegetation restoration, and water resource management.
**Keywords:** the coupled carbon and water (CCW) model; runoff change; climate change;
vegetation change; increasing atmospheric $CO_2$ concentrations; attribution analysis



**Plain language:** Climate change, rising $CO_2$, and vegetation dynamics are reshaping
global water cycle, but their impacts remain unclear. We improved the coupled carbon
and water model to analyze China's water yield (WY) changes (1982–2017). Our
results showed that climate change was the dominant driver nationally, vegetation/ $CO_2$
most affected in 400-1600 mm precipitation zones. Projections indicate $CO_2$ may
increase WY 1.3% annually by 2100, surpassing other drivers. This work informs
sustainable water management.





## 1  Introduction


The global environment has been undergoing rapid changes, impacting
hydrological processes through climate change, escalating atmospheric $CO_2$
concentration [$CO_2$], and vegetation dynamics. Notably, China has experienced a
visible greening trend in recent decades, prompting a heightened focus on ecological
and water resource concerns (Chen et al., 2019). Investigating the influence of
vegetation changes on runoff has thus emerged as a pivotal research area, aligning with
China's increasing emphasis on environmental sustainability. Understanding the
intricate interplay among vegetation dynamics, climate change, and [$CO_2$] within the
water cycle, particularly concerning runoff, holds significant promise for informing
future water resource management strategies and ecosystem preservation initiatives and
offering valuable insights for climate change adaptation endeavors (Ogutu et al., 2021;
Yang et al., 2019).
Climate change directly affects runoff by altering precipitation patterns,
temperature regimes, and radiation levels (Ban et al., 2023; Li et al., 2022). It also
indirectly influences runoff dynamics by altering vegetation phenology (Liu et al.,
2024). Vegetation, in turn, plays a key role in the hydrological cycle by influencing root
water uptake, canopy transpiration, rainfall interception, and soil infiltration processes
(Hoek Van Dijke et al., 2022; Shi et al., 2022; Yang et al., 2023; Zhang et al., 2022a).
Additionally, rising [$CO_2$] affects transpiration by influencing vegetation
photosynthesis, thus indirectly impacting hydrological processes (Wei et al., 2024;
Zhou et al., 2023). Although recent studies have attempted to separate the impacts of
vegetation from climate using ecohydrological models, the results remain inconsistent
(Fu et al., 2023; Yang et al., 2020). Some research suggested that climate change had a
more significant direct impact on runoff (Liu et al., 2024; Yang et al., 2021; Zhai and
Tao, 2017, 2021), while others highlighted the comparable or even dominant role of
vegetation change and [$CO_2$] in runoff dynamics (Li et al., 2020b; Wang et al., 2021;





Zhou et al., 2023). Therefore, further research is needed to disentangle the complex
effects of climate, vegetation, and [$CO_2$] change on runoff.
Several methods have been employed to separate the effects of climate, vegetation,
and [$CO_2$] change on runoff change, including paired catchment experiments, statistical
methods, and modeling approaches (Zeng et al., 2020). Given that annual water yield
(WY) equates to runoff through negligible soil water storage changes, these
methodological evaluations directly inform WY attribution frameworks (Zhang et al.,
2022c). The paired catchment experiment method, though classical, is limited to small-
scale watersheds and is less applicable to larger regions (Peng et al., 2016). Statistical
methods, while helpful in identifying correlations, lack a physical basis and are
insufficient for explaining the underlying mechanisms of runoff changes (Chen et al.,
2022). Modeling approaches, which are broadly categorized into conventional
hydrological models and ecohydrological models, provide a more systematic
framework for attribution analysis. Conventional hydrological models tend to focus on
runoff simulation and often oversimplify the effects of vegetation and [$CO_2$],
potentially underestimating their impacts on runoff (Zhai and Tao, 2021).
Ecohydrological models, which consider both hydrological and vegetation processes,
can better separate the effects of climate, vegetation, and [$CO_2$], but are often
computationally demanding and limited in their spatial applicability (Jiao et al., 2017;
Ma et al., 2023). Among these modeling approaches, the Budyko framework, widely
used to separate climate change effects on runoff, quantifies water balance through the
aridity index (PET/precipitation) and incorporates a catchment-specific parameter "n"
representing integrated land surface characteristics (e.g., vegetation, soil, topography)
(Zhang et al., 2022b, 2016a). However, existing studies typically attributed temporal
changes in "n" solely to vegetation change (Tan et al., 2024; Xue et al., 2022; Zhou et
al., 2023) or correlated "n" with vegetation indices (e.g., NDVI) through multivariate
regression (Liu et al., 2024; Tan et al., 2023)—might not accurately reflect the true
impact of vegetation change. This is because the approach oversimplifies the role of "n"



by conflating vegetation effects with confounding factors (e.g., $CO_2$-induced stomatal
adjustments, and climate change), as regression-based methods inherently fail to
disentangle covarying drivers, thereby obscuring whether "n" changes originate from
vegetation dynamics, $CO_2$-physiological feedbacks, or multi-factor synergies (Gan et
al., 2021). While some studies incorporated [$CO_2$] effects via PET adjustments instead
of actual evapotranspiration, this indirect approach conflates $CO_2$-driven PET changes
with other meteorological drivers (e.g., radiation, wind) and propagates parameter
uncertainties (e.g., "n"), obscuring [$CO_2$]'s independent impact on runoff (Liu et al.,

2024).

The coupled carbon and water (CCW) model integrates hydrological and

ecological processes by mechanistically linking vegetation dynamics to water and
carbon fluxes through remote sensing-driven parameterization (Li et al., 2024b; Zhang
et al., 2021b, 2022c). Unlike the Budyko framework's empirical parameter "n"—which
conflates vegetation effects with unaccounted catchment characteristics—CCW
explicitly resolves vegetation impacts through two distinct pathways: (1) structural
effects—quantified by NDVI-modulated canopy absorption of photosynthetically
active radiation (FPAR) that captures changes in energy partitioning due to vegetation
greening; and (2) physiological adjustments—represented by biome-specific variations
in underlying water-use efficiency (UWUE) and vapor pressure deficit (VPD)-
mediated regulation of evapotranspiration (ET). In the model, GPP is estimated from
light-use efficiency theory ($\varepsilon_{pot} \times$ FPAR $\times$ PAR$\times$ Rs $\times$ Ts $\times$ Ws), and ET is
mechanistically coupled to GPP via UWUE—a physiologically grounded parameter
representing ecosystem-level carbon–water trade-offs, calibrated against global
FLUXNET observations (Zhang et al., 2016b), which encapsulates ecosystem-level
carbon–water trade-offs. By contrast, Budyko's empirical "n" aggregates these distinct
vegetation controls into a single catchment-scale parameter, obscuring their individual
hydrological impacts.





Nevertheless, the original CCW model, while robust in capturing vegetation-
climate interactions, does not account for $CO_2$-induced physiological changes,
specifically long-term enhancements in water-use efficiency (WUE) resulting from
elevated [$CO_2$] (Adams et al., 2020; Li et al., 2023). This omission limits its ability to
isolate [$CO_2$] fertilization effects from climate and LULC (land use and land cover)
changes, a gap particularly problematic in regions like China, where $CO_2$-driven WUE
improvements may offset or amplify vegetation greening impacts on runoff.
Therefore, we aim to improve the CCW model by incorporating dynamic WUE
responses to [$CO_2$], building on the biome-specific UWUE framework. Furthermore,
by integrating $CO_2$-dependent WUE adjustments into the ET-GPP coupling, our
improved model explicitly partitions runoff changes into three causal drivers: (1)
climate change (eg. precipitation, temperature, and so on), (2) vegetation structural
changes (NDVI, and land use and land cover (LULC)), and (3) $CO_2$-physiological
effects (stomatal optimization).

## 2    Methods and Data

### 2.1    Data sources and processing

Four main datasets were employed in the improved CCW model: vegetation data
(NDVI), climate data (precipitation, temperature, shortwave radiation, vapor pressure
deficit, and atmospheric pressure), land use and land cover (LULC), and [$CO_2$]. The
monthly NDVI dataset used in this study (Table 1) was derived from a daily 0.05° gap-
free NDVI dataset in China (https://doi.org/10.6084/m9.figshare.c.7002225.v1) (Li et
al., 2024a), which was developed from the NOAA's daily NDVI dataset, applying
effective data recognition and spatiotemporal gap-filling techniques. The dataset spans
1981–2023 and provides a spatial resolution of 0.05°, and we used bilinear interpolation
to generate the dataset with a spatial resolution of 0.1°.
Climate data (Table 1), including precipitation, air temperature, surface downward
shortwave radiation, relative humidity, and atmospheric pressure, were sourced from





the China Meteorological Forcing Dataset (CMFD) at the National Tibetan Plateau
Data Center (TPDC) of the Institute of Tibetan Plateau Research, Chinese Academy of
Sciences (He et al., 2020). The dataset spans 1979–2018 and provides a spatial
resolution of 0.1° and temporal resolutions at 3-hour, daily, monthly, and annual scales.
As the dataset did not provide vapor pressure deficit (VPD), we calculated VPD using
the method from Howell and Dusek (1995), based on atmospheric pressure, temperature,
and relative humidity.

LULC data (Table 1) were obtained from the Zhang et al. (2024) global dataset,

which provides consistent multi-temporal global LULC maps at 30 m spatial resolution
for 1985–2022. The dataset includes 35 fine-resolution LULC types. For the purposes
of this study, and to facilitate LULC change analysis, we merged these 35 LULC types
into 17 types using the IGBP classification, based on the method by Yang et al. (2017).
Four primary LULC types—cropland, forest, grassland, and bare land—were
determined following the method described by Mu et al. (2013). The data were
resampled to the 0.1° spatial resolution, ensuring compatibility for modeling within the
modified CCW framework.

[CO$_2$] data were sourced from the Mauna Loa Observatory (MLO), Hawaii (20°N,

156°W)    (http://cdiac.esd.ornl.gov/ftp/trends/co2/    maunaloa.co2),    with    yearly
observations used to represent national [CO$_2$] levels due to the minimal spatial variation
in [CO$_2$] across China (Table 1). These datasets were then used to drive the improved
CCW model.

In this study, the hydrological data for model validation from 145 hydrological

stations (Fig. 1), each with at least 15 years of continuous data since 1982, was collected
from the Hydrological Bureau of the Ministry of Water Resources of China
(https://www.mwr.gov.cn/english/). Annual runoff data were calculated from the daily
runoff and the catchment area controlled by each hydrological station.
**Table 1**. Hydrology, climate, and vegetation data for the improved CCW model



| Dataset | Original Resolution (spatial/temporal) | Period | Reference |
|---------|----------------------------------------|--------|-----------|
| NDVI | $0.05° \times 0.05°$ (daily) | 1981 - 2023 | (Li et al., 2024a) |
| Landcover | 30m×30m (5-year) | 1985 - 2022 | (Zhang et al., 2024) |
| Climate | $0.1° \times 0.1°$ (monthly) | 1979 - 2018 | (He et al., 2020) |
| [CO$_2$] | yearly | 1959 - 2023 | Mauna Loa Observatory, Hawaii |
| Streamflow | daily | 1982 - 1995 (or later) | On-site streamflow records and the regional flow summary reports of government |

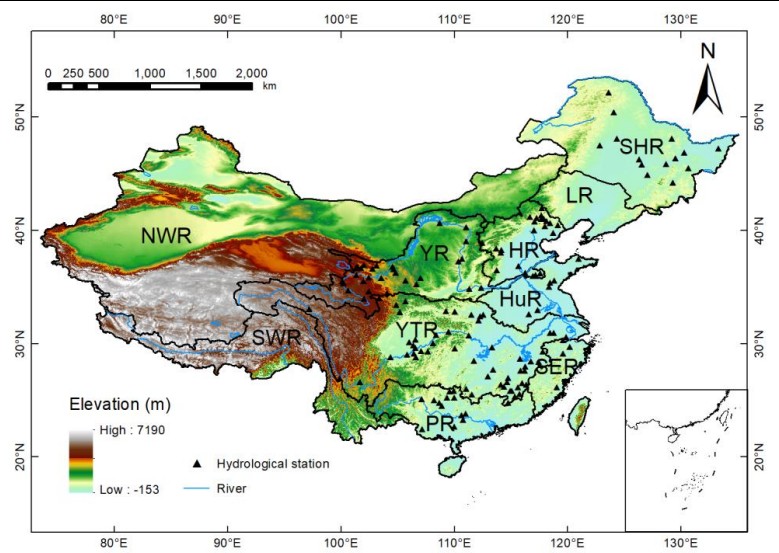

**Figure 1.** The geographic location and topography of the study area, where the black triangles mark the location of the hydrological gauging stations for model evaluation. Ten river basins considered in this study are: Songhua River basin (SHR), Liao River basin (LR), Hai River basin (HR), Huai River basin (HuR), Yangtze River basin (YZR), Yellow River basin (YR), Pearl River basin (PR), Southeast Rivers (SER), Southwest Rivers (SWR) and Northwest Rivers (NWR).

## 2.2 The improved CCW model

The original Coupled Carbon and Water (CCW) model (Zhang et al., 2016b) is a data-driven, remote sensing-based model that effectively integrates carbon and water dynamics to estimate monthly gross primary productivity (GPP) and evapotranspiration (ET). This model, which is particularly carbon-centric, derives ET from GPP constrained by underlying water-use efficiency (UWUE) parameters, which were calibrated using global FLUXNET data (Zhang et al., 2016b; Zhou et al., 2014). Despite



its simpler structure, the CCW model achieves accuracy comparable to more complex
process-based models in ET estimation. The essential components of the CCW model
are represented as:
$$GPP = APAR \times \varepsilon = PAR \times FPAR \times \varepsilon_{pot} \times R_s \times T_s \times W_s \qquad (1)$$
where APAR is the absorbed photosynthetically active radiation (MJ m$^{-2}$), which is
calculated as the product of incident photosynthetically active radiation (PAR) and the
fraction of PAR absorbed by vegetation (FPAR), and PAR is typically assumed to be
45% of the total shortwave radiation (Running et al., 2000); FPAR is determined by the
normalized difference vegetation index (NDVI) (Sims et al., 2005); $\varepsilon$ is the realized
light-use efficiency (g C MJ$^{-1}$), which is calculated by multiplying the potential light-
use efficiency ($\varepsilon_{pot}$) and environmental scalars for diffuse radiation (Rs), temperature
(Ts), and moisture stress (Ws). This formulation ensures that GPP estimates reflect the
influence of radiation, temperature, and moisture limitations on photosynthetic activity.

In this study, we improve the CCW model by incorporating dynamic water use

efficiency (WUE) instead of static UWUE. This enhancement addresses the limitations
of the original model, particularly its inability to adapt to environmental changes such
as varying [CO$_2$] and vapor pressure deficit (VPD). WUE's estimation method is
estimated using the WEC (Water Efficiency and Carbon) equation proposed by Cheng
et al. (2017). The final formula for calculating WUE is:
$$WUE = \frac{C_a \times P_a}{1.6(VPD + g_1\sqrt{VPD})}[1 - exp(-k * LAI)](1 - f_i) \qquad (2)$$
where $C_a$ is atmospheric CO$_2$ concentration (mol(CO$_2$) mol$^{-1}$(air)); Pa is atmospheric
pressure (kPa); VPD is vapor pressure deficit (kPa); g$_1$ is an empirical parameter of the
Ball stomatal conductance model; k is the radiation extinction coefficient, typically set
at 0.6, describing how light is absorbed by the canopy; LAI is the leaf area index; and
$f_i$ is a factor representing nonproductive water use (such as evaporation from soil and
canopy interception). This equation provides a dynamic estimate of WUE, considering
the effects of environmental factors like VPD, CO$_2$ concentration, atmospheric pressure,





and canopy structure (LAI). The factor 1−exp(−k×LAI) accounts for light interception
by the canopy.

In order to ensure the consistency of NDVI and LAI trends, we calculated LAI

using NDVI (Gutman and Ignatov, 1998) instead of LAI dataset:
$$
\begin{cases}
LAI = -2ln(1 - f_{NDVI}) \\
f_{NDVI} = \dfrac{NDVI - NDVI_0}{NDVI_1 - NDVI_0}
\end{cases}
\tag{3}
$$

where $NDVI_0 = 0.04$, $NDVI_1 = 0.52$

Evapotranspiration (ET) is then calculated as the ratio of GPP to WUE:

$$
ET = \frac{GPP}{WUE}
\tag{4}
$$

This modification allows the model to estimate ET using dynamic WUE, replacing the
static UWUE from the original model. The dynamic nature of WUE enhances the
model's ability to simulate ecosystem water use across different environmental
conditions and vegetation types.

Finally, the water yield (WY) is calculated as the difference between precipitation

(P) and ET:
$$
WY = P - ET
\tag{5}
$$

On an annual scale, WY is assumed to be approximately equal to runoff, as

changes in soil water storage over long periods (one year or longer) are considered
negligible. Thus, the attribution of WY can also be considered as the attribution of
runoff.
**2.3  Attribution analysis framework**

To explore the combined and individual effects of climate, vegetation, and [$CO_2$]

change on water yield (WY), four scenarios were designed based on data from 1982 to
2017 (Table 2). Scenario 1 (Actual) aimed to validate the improved CCW model and
estimate the combined effects of climate, vegetation, and [$CO_2$] change on WY by





allowing all variables to vary from 1982 to 2017. Scenario 2 (Vegetation Change)
focused on estimating the direct effects of vegetation change on WY by allowing
vegetation variables (NDVI and LULC) to vary while keeping climate and [$CO_2$] fixed
at 1982 levels. In this case, the trend in WY obtained reflects the impact of vegetation
change alone. Scenario 3 (Climate Change) aimed to estimate the direct effects of
climate change on WY by allowing climate variables (precipitation, temperature,
relative humidity, solar radiation, and atmospheric pressure) to change, while fixing
vegetation and [$CO_2$] at 1982 levels. This scenario helps isolate the effects of climate
change on WY. Scenario 4 ([$CO_2$] Change) was designed to estimate the direct effects
of [$CO_2$] change on WY by varying [$CO_2$] levels from 1982 to 2017, while climate and
vegetation variables were fixed at 1982 levels. The resulting WY trend reflects the
impact of [$CO_2$] change alone.
**Table 2**. Scenario designs in the improved CCW model for WY attribution. LULC: Land use
and land cover types; NDVI: Normalized difference vegetation index; TMP: Temperature;
SRAD: Shortwave radiation; VPD: Vapor pressure deficit.

| Scenarios | Vegetation | | Climate | | | | | $CO_2$ | Purposes |
|---|---|---|---|---|---|---|---|---|---|
| | LULC | NDVI | P | T | RH | Srad | Pa | $CO_2$ | |
| S1 (baseline) | ▲ | ▲ | ▲ | ▲ | ▲ | ▲ | ▲ | ▲ | Validating the improved CCW model and estimating the combined effects of climate, vegetation, and $CO_2$ change. |
| S2 (vegetation) | ▲ | ▲ | △ | △ | △ | △ | △ | △ | Estimating the direct effects of vegetation change. |
| S3 (climate) | △ | △ | ▲ | ▲ | ▲ | ▲ | ▲ | △ | Estimating the direct effects of climate change. |





| S4 (CO₂) | △ | △ | △ | △ | △ | △ | △ | ▲ | Estimating the direct effects of CO₂ change. |
|---|---|---|---|---|---|---|---|---|---|

Note: The symbol "▲" denotes a changing input variable over time, whereas the symbol "△"
represents a fixed input variable at the level of the initial year (1982).
The relative contributions of climate, vegetation, and [CO₂] to changes in WY
were calculated using the following formula:

$$
\begin{cases}
RC_{vegetation} = \dfrac{trend_{vegetation}}{\left|trend_{vegetation}\right| + \left|trend_{climate}\right| + \left|trend_{CO2}\right|} \times 100\% \\[2ex]
RC_{climate} = \dfrac{trend_{climate}}{\left|trend_{vegetation}\right| + \left|trend_{climate}\right| + \left|trend_{CO2}\right|} \times 100\% \\[2ex]
RC_{CO2} = \dfrac{trend_{CO2}}{\left|trend_{vegetation}\right| + \left|trend_{climate}\right| + \left|trend_{CO2}\right|} \times 100\%
\end{cases}
\tag{6}
$$

where $trend_{vegetation}$, $trend_{climate}$, and $trend_{CO2}$ represent the changes in water
yield (WY) resulting from vegetation, climate, and [CO₂] changes, respectively, as
calculated in each scenario; the relative contributions ($RC_{vegetation}$, $RC_{climate}$, and
$RC_{CO2}$) are expressed as percentages, indicating the proportion of each factor's
influence on the overall changes in WY.
At each grid point, the absolute values of the relative contributions of each factor
(vegetation, climate, and [CO₂]) are compared. For each grid point, we identify the
most significant contributor to water yield (WY) changes by comparing the relative
contributions of each factor. If the absolute values of the relative contributions of two
factors do not exceed 5%, then these two factors are considered joint significant
contributors to the changes in WY at that grid point (Ma et al., 2024; Saltelli et al.,
2007). This approach helps to highlight areas where the impacts of multiple factors are
closely intertwined and both play a critical role in influencing water yield, suggesting
that their combined effects are comparable in magnitude. In these cases, the relative
contribution of each factor is not significantly stronger than the other, indicating that
their combined influence on WY is equally important at the local scale.



The scenario analysis previously conducted revealed the relative contributions of
climate, vegetation, and [$CO_2$] to WY changes. However, these contributions arise from
both the intrinsic rate of change of each factor and the sensitivity of runoff to those
changes (the elasticity coefficient) (Yang and Yang, 2011). To gain a deeper
understanding of the changes in WY, we employ elasticity coefficients to quantify its
sensitivity to individual factor. We specifically focused on precipitation because,
despite not always having the highest sensitivity, it is integral to the hydrological cycle
and essential for assessing water yield (WY) under various climate change scenarios
(Liu et al., 2017). The elasticity of runoff refers to the variation in runoff depth resulting
from a 1% increase in each climatic variable (Xu et al., 2014). The absolute value of
elasticity reflects the sensitivity of runoff to various influencing factors. In other
methods, elasticity coefficients are typically calculated using an analytical expression
based on instantaneous changes in runoff corresponding to variations in a given factor
in a specific year (Fu et al., 2023; Liu et al., 2017; Yang and Yang, 2012). However, in
our study, we applied scenario-based analysis over the period of 1982 to 2017. This
extended temporal window allowed us to better account for the long-term effects and
interactions of multiple factors influencing WY. So we vary each factor (precipitation,
NDVI, and [$CO_2$]) by 1% relative to the baseline scenario S1 across the entire 1982-
2017 period. We then calculated the annual average runoff values from the adjusted
sequence and compared them with the average original baseline runoff values. The
difference between these two values, divided by the average baseline runoff value, gave
us the runoff change rate:
$$\frac{\Delta R_x}{R_x} = \frac{WY_{mean_x} - WY_{mean_x}}{WY_{mean_x}} \tag{7}$$

Mathematically, the elasticity coefficient is defined as the runoff change rate
divided by 1%, and the formula is as follows:
$$\varepsilon_x = \frac{\frac{\Delta R_x}{R_x}}{\frac{\Delta x}{x}} = \frac{\frac{\Delta R_x}{R_x}}{1\%} \tag{8}$$





Generally, while the scenario analysis above has identified which factors are most
influential based on their relative contributions, the elasticity coefficients allow us to
explain why these factors are critical by demonstrating their respective impacts on WY
through sensitivity analysis. This dual approach—combining both the changes in the
factors and their elasticities—provides a more comprehensive understanding of the
drivers behind the observed changes in WY, ensuring that the results of the scenario
analysis are both meaningful and robust.
**3    Results**
**3.1    Changes in hydrometeorological variables**
Fig. 2 demonstrates the trends of annual precipitation, air temperature, relative
humidity, atmospheric pressure, solar radiation, and NDVI across China during 1982-
2017. Annual precipitation change exhibited a clear spatial distribution pattern,
specifically decreases in central China, including the middle reaches of the Yellow
River and the Yangtze River basins, and increases in the northwest and southeast. Air
temperature exhibited a consistent warming trend across China. In contrast, relative
humidity generally decreased across most China. Atmospheric pressure remained
relatively stable. Regarding solar radiation, decreases were in northern China, while an
increase was in southern regions. The decreasing solar radiation in northern China is
likely due to increased aerosol concentrations (Liang et al., 2024). NDVI showed a
significant increasing trend, which indicates an overall enhancement in vegetation
growth across China. This trend was especially prominent in central and eastern regions,
including the Yellow River Basin and the Yangtze River Basin. In these regions, LULC
changes, such as afforestation and agricultural practices, likely contributed to the
observed increases in NDVI (Chen et al., 2019).

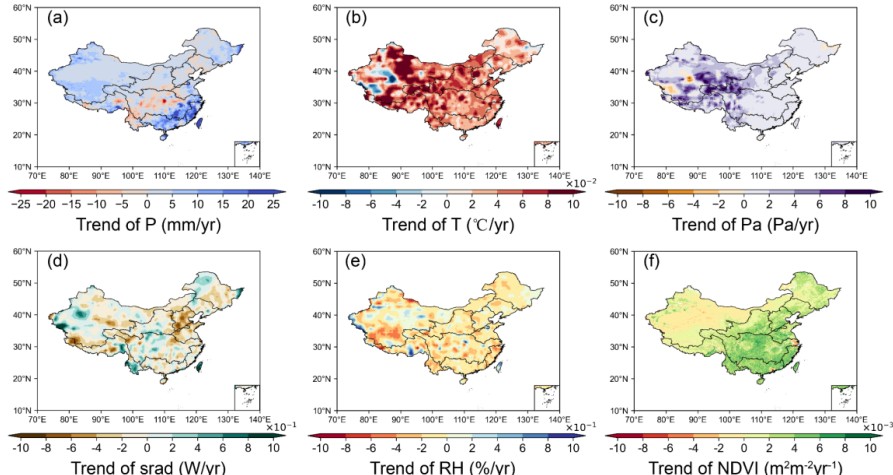

**Figure 2.** Spatial patterns of trends in annual climatic and vegetation variables during 1982–2017. (a) precipitation (mm/yr); (b) air temperature (°C/yr); (c) Atmospheric pressure (Pa/yr); (d) shortwave radiation (W/m²/yr); (e) relative humidity (%/yr); (f) NDVI (yr$^{-1}$).

Significant changes in land use and land cover (LULC) occurred in China during 1982-2017, as illustrated in Fig. 3. Although the overall percentage distribution of major land cover types, namely grasslands, forests, croplands, and bare lands, remained relatively stable, these four categories dominated the landscape, with most changes concentrated within them. Notably, the transitions among these categories were characterized by mutual conversions, particularly from bare land to grasslands (Fig. 3). Spatially, the changes exhibited distinct regional patterns. In southern China, LULC changes were mainly characterized by the conversion of land to forests and grasslands. In contrast, the northeastern regions exhibited more complex transformations, with some areas shifting to bare land and croplands (Fig. 3).

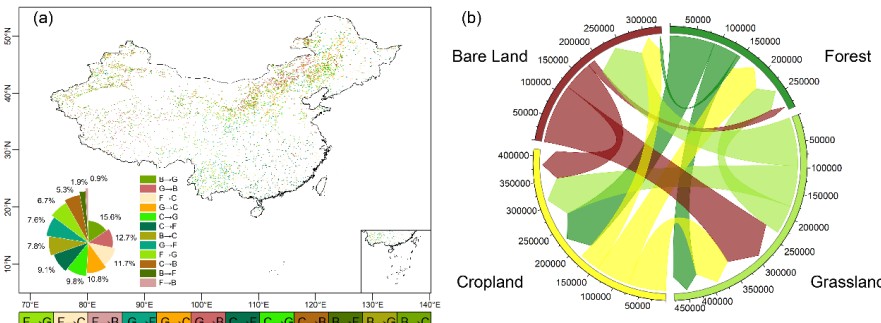

**Figure 3.** Land use and land cover (LULC) changes from 1982 to 2017. (a) Spatial pattern distribution of LULC change; (b) Chord diagram of LULC conversion flows (unit: km$^2$), where directional arrows represent transitions between land types (originating type → current type), with chord widths proportional to the converted areas. The figure illustrates the converted areas and does not include the unchanged regions.

## 3.2   Performance of the improved CCW model

As shown in Fig. 4a, the observed  annual water yield (WY) and the simulated annual WY by the improved CCW model showed strong linear correlations ($R^2 = 0.7$), with the regression line slope being 1.45, $R^2$ being 0.7, and RMSE being 12.49 mm/year. It indicates that the model provides a reliable representation of the observed trends.

The estimated annual WY trends had distinct spatial patterns (Fig. 4b), which closely aligned with that of precipitation. Specifically, decrease trends in WY occurred in the central regions of the Yellow River Basin and the middle section of the Yangtze River Basin, while increase trends were found in other regions, with the southeast exhibiting the highest rate of increase.



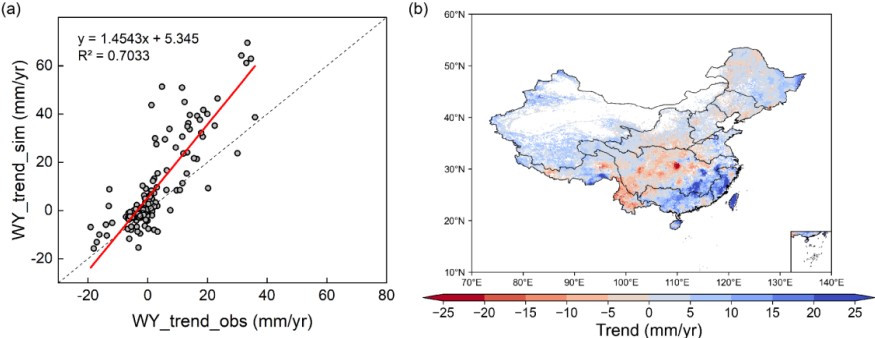

**Figure 4.** (a) Validation of simulated WY trend using the improved CCW model; (b) Spatial distribution of WY trends under scenario S1(actual situation) during 1982–2017.

## 3.3 Attribution analysis of annual WY changes

Fig. 5 shows the distribution of WY changes caused by climate, vegetation, and [$CO_2$] changes, integrating both absolute magnitude (Fig. 5a-c) and relative dominance (Fig. 5d-f) of their contributions. Climate-driven WY changes exhibited marked spatial heterogeneity, with absolute increases exceeding 15 mm/yr in southeastern China (Fig. 5a), corresponding to 60-90% relative contributions (Fig. 5d). Central basins showed contrasting declines of 0-6 mm/yr under climate forcing, while northeastern transitional zones displayed mixed positive/negative absolute changes (Fig. 5a) despite maintaining 40-70% relative climate dominance (Fig. 5d). This spatial heterogeneity aligned with precipitation change patterns (Fig. 2a).





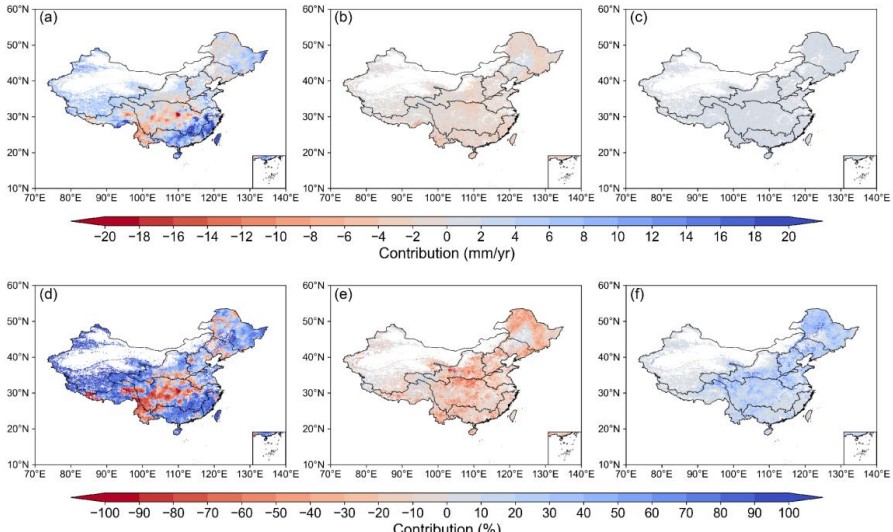

**Figure 5.** The absolute contributions of (a) climate, (b) vegetation, and (c) [$CO_2$], and the relative contributions of (d) climate, (e) vegetation, and (f) [$CO_2$] to changes in WY trends for 1982-2017.

Vegetation-mediated WY reductions reached 0-6 mm/yr (Fig. 5b), accompanied by 0-60% relative contributions (Fig. 5e). These effects originated from enhanced evapotranspiration through land-use changes and NDVI-based greening, particularly pronounced in central China. Specific regions in the Yangtze, Yellow, and northeastern rivers showed vegetation-driven relative contributions reaching 40-60% (Fig. 5e). [$CO_2$] effects generated limited direct absolute impacts (<5 mm/yr, Fig. 5c) but exerted 10-40% relative influences (Fig. 5f) through stomatal closure mechanisms. This process partially counteracted vegetation-related WY losses in transitional climates like northeastern China, where competing drivers created complex ecohydrological interactions (Fig. 5d-f).

Fig. 6 illustrated the spatial distribution of WY trend drivers over the past four decades. Climate change was the dominant factor of WY variation in more than 70% regions, mainly in the Northwest, Southwest, Southeast, Pearl River basins, and other parts of the Yangtze and Yellow River basins. Vegetation changes ranked as the secondary control, dominating WY changes in parts of the Yangtze, Yellow, Songhua,



Liao, and Hai Rivers. Remarkably, it was shown that the region where vegetation and
[$CO_2$] had the dominant influence mainly distributes within precipitation ranges of 400–
1600 mm. $CO_2$-induced effects were least influential at a national scale. This three-
tiered hierarchy—climate changes as the primary forcing, vegetation changes as the
secondary control, and [$CO_2$] effects as a localized modifier—reveals how hydrological
regimes govern the spatial succession of dominant drivers across China's diverse
ecohydrological gradients.

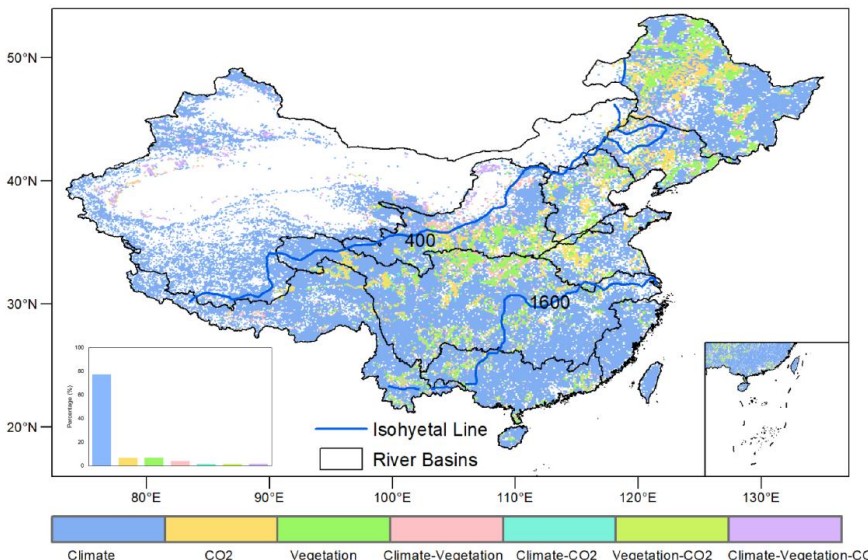


**Figure 6.** Spatial distributions of dominant factors controlling WY change. Driving factors
include climate, vegetation, and [$CO_2$]. Climate: Areas where climate (e.g., precipitation,
temperature) is the dominant factor influencing WY change; CO2: Areas where [$CO_2$]
is the primary driver of WY change; Vegetation: Areas where vegetation changes (e.g.,
NDVI, LULC) primarily drive WY changes. Climate-Vegetation: Areas where both climate
and vegetation jointly influence WY; Climate-CO2: Areas where both climate and [$CO_2$] jointly
contribute to WY change; Vegetation-CO2: Areas where vegetation changes and [$CO_2$] jointly
control WY; Climate-Vegetation-CO2: Areas where the combined effect of climate, vegetation,
and [$CO_2$] jointly controls WY change. Additionally, the approximate isohyetal line shown in
the figure were derived based on annual precipitation data from 1982 to 2017.





### 3.4 Elasticity of WY to main variables


The sensitivity of WY to precipitation ($\varepsilon P$), NDVI ($\varepsilon NDVI$), and [$CO_2$] ($\varepsilon CO_2$)
exhibits distinct spatial patterns in (Fig. 7). Nationally averaged elasticity coefficients
showed that a 10% increase in precipitation, [$CO_2$], and NDVI altered WY by 15.5%
($\varepsilon P$=1.55), 5.5% ($\varepsilon CO_2$=0.55), and -4.4% ($\varepsilon NDVI$=-0.44), respectively, indicating that,
in terms of the sensitivity of runoff to changes in each factor, the ranking was
precipitation > [$CO_2$] > NDVI.

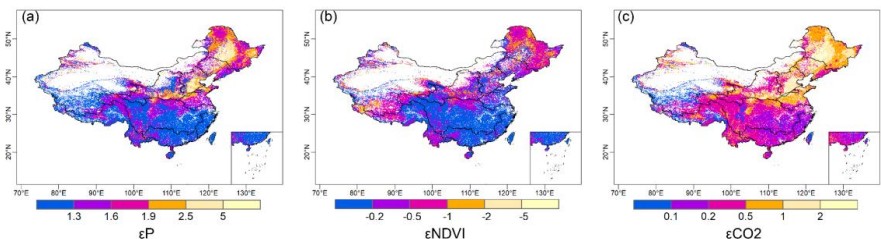

**Figure 7. S**patial distribution of elasticity coefficients of WY relative to changes in
hydrological variables such as (a) annual precipitation, (b) NDVI, and (c) [$CO_2$].
The elasticity coefficients of precipitation ($\varepsilon P$), [$CO_2$] ($\varepsilon CO_2$), and vegetation
($|\varepsilon NDVI|$) all exhibited a coherent latitudinal decline across China's river basins,
showing systematically higher sensitivity in northern regions than southern
counterparts. Quantitatively, $\varepsilon P$ decreased from 2.09 in the Songhua River basin to 1.15
in the Southeastern Basin, accompanied by similar reductions in $|\varepsilon NDVI|$ (from 0.76 to
0.13) and $\varepsilon CO_2$ (from 1.08 to 0.16) (Table 3).
A distinct abrupt transition zone in elasticity coefficients was identified around
33°N, closely aligning with China's traditional North-South physiographic divide.
Around the zone, elasticity coefficients exhibited an abrupt decline from the Yellow
River Basin to the Yangtze River Basin. Specifically, the Yellow River Basin showed
higher sensitivities to precipitation ($\varepsilon P$=1.87), [$CO_2$] ($\varepsilon CO_2$=0.86), and NDVI
($\varepsilon NDVI$=-0.53), which were approximately 1.4, 2.8, and 2.8 times greater, respectively,
than those in the Yangtze River Basin ($\varepsilon P$=1.31, $\varepsilon CO_2$=0.31, $\varepsilon NDVI$=-0.19).





**Table 3**. Elasticity Coefficients of Runoff to Precipitation, NDVI, and $CO_2$ in Different
Watersheds

| Dataset | $\varepsilon P$ | $\varepsilon NDVI$ | $\varepsilon CO_2$ |
|---|---|---|---|
| **Songhua River basin** | 2.09 | -0.76 | 1.08 |
| **Hai River basin** | 2.13 | -0.44 | 1.12 |
| **Yellow River basin** | 1.87 | -0.53 | 0.86 |
| **Yangtze River Basin** | 1.31 | -0.19 | 0.31 |
| **Huai River basin** | 1.64 | -0.18 | 0.63 |
| **Pearl River basin** | 1.25 | -0.17 | 0.25 |
| **Southeast Rivers** | 1.15 | -0.13 | 0.15 |

Note: Some LULC types were excluded from the analysis. Due to many missing data points,
the Liao River, Southwest, and Northwest river basins were also omitted.
## 4  Discussion
### 4.1  Strength of the attribution analysis framework
To address limitations in current methods for analysing the effects of climate,
vegetation, and [$CO_2$] on runoff changes, we developed an attribution analysis
framework based on the improved CCW model. This framework has been improved in
three aspects. Firstly, the explicit and mechanistic integration of vegetation dynamics
and [$CO_2$] effects overcomes the oversimplifications inherent in conventional
approaches. Traditional Budyko-based frameworks often attribute vegetation effects to
temporal variations in the parameter "n" by either statistically regressing "n" against
vegetation proxies such as NDVI  (Liu et al., 2024; Tan et al., 2023) or simplistically
equating "n" to vegetation effects (Li et al., 2020b; Zhou et al., 2023).  Such approaches
conflate structural vegetation changes (e.g., leaf area index) with physiological
adjustments (e.g., $CO_2$-induced stomatal closure), thereby obscuring the independent
roles of vegetation dynamics and [$CO_2$]. For example, while rising [$CO_2$] levels directly
reduce stomatal conductance and transpiration, Budyko-based studies often
misinterpret this effect as part of the "n" parameter's variability, erroneously attributing
it to vegetation changes (Zeng et al., 2020). In contrast, our framework mechanistically
separates these pathways: structural modifications are distinguished from $CO_2$-driven





stomatal physiological responses, resolving contradictions in prior findings where
vegetation greening was reported to both mitigate (Zeng et al., 2018) and exacerbate
(Farley et al., 2005) runoff changes.

Secondly, unlike Budyko-based methods that indirectly represent $[CO_2]$ impacts

through adjustments to potential evapotranspiration (PET)—a practice conflating $[CO_2]$
effects with meteorological drivers like radiation and wind—our framework explicitly
quantifies $CO_2$'s physiological influence on actual evapotranspiration (AET) by
mechanistically modeling its role in stomatal conductance and water-use efficiency
(WUE). Elevated $[CO_2]$ reduces stomatal aperture, directly suppressing transpiration
while enhancing carbon assimilation. For example, our results show that reduction in
transpiration due to $CO_2$-driven stomatal closure offsets water losses, a mechanism
entirely masked in Budyko frameworks where $[CO_2]$ effects are ambiguously
embedded in PET adjustments or erroneously attributed to vegetation structural
changes via the "n" parameter (Liu et al., 2024).

Thirdly, while numerous studies have conducted runoff attribution analysis at the

basin scale (Liu et al., 2024, 2017; Yang et al., 2022), our grid-scale approach
transcends the spatial constraints of fixed watershed boundaries by resolving regional
heterogeneity in hydrological drivers. Conventional basin-aggregated methods obscure
critical intra-basin differences—for instance, our analysis reveals that grids in the upper
Yangtze River basin, where precipitation change dominates runoff trends, necessitate
climate scenario-based water resource planning. In contrast, mid-basin grids with
significant NDVI-driven greening exhibit pronounced WY reductions, highlighting the
need for vegetation management strategies that restrict excessive afforestation in water-
sensitive areas (Sun et al., 2022; Yang et al., 2021). By decoupling analysis from rigid
watershed boundaries, our framework enables targeted strategies such as restricting
reforestation in water-stressed grids or selecting $CO_2$-adapted vegetation species,
thereby aligning management actions with localized climate-vegetation-hydrology
interactions.



## 4.2 New insights into attribution analysis

Our findings highlighted climate change as the dominant driver of water yield (WY) changes (contributing >70%), consistent with other assessments (Table 4), yet reveal critical regional divergences. Climate impacts dominated in the Northwest and Southwest River Basins, as well as parts of the Yangtze, Yellow, Southeast, and Pearl River Basins, while vegetation and $[CO_2]$ effects prevailed in central China (parts of the Yangtze, Yellow, Songhua, Liao, and Hai River basin)—a spatial pattern slightly distinct from earlier studies. Although previous studies identified human activities as the primary driver in some northern basins (Liao, Hai, and Yellow River Basins) (Yang et al., 2022), their long-term study (1965-2018) diluted the gradually strengthening vegetation signals after 2000 mentioned in other studies (Liu et al., 2017; Sun et al., 2023) through time-averaging. Our findings now confirm the emerging importance of vegetation dynamics in southern basins like the Yangtze through our symmetric 1982-2017 study period.

**Table 4.** Comparative studies of the contribution of climate variability and vegetation to runoff changes.

| Reference | Study region | Study period | Method/Model | Driving factors |
|---|---|---|---|---|
| **(Wei et al., 2024)** | Global | 1981-2020 | Trendy phase 11 +ROF | Climate change |
| **(Liu et al., 2024)** | Global | 1984-2010; 2000-2100 | Improved Budyko | Precipitation |
| **(Zhou et al., 2023)** | Global | 1850-2014; 2015-2100 | Improved Budyko + CMIP6 | Land surface changes |
| **(Tan et al., 2023)** | Global | 2003-2016; 1982-2016 | Improved Budyko | Effective precipitation |
| **(Yang et al., 2022)** | China | 1965-2018 | Budyko | P: Northwest river basin, Southwest river basin, Yangtze river basin, Southeast river basin, and Pearl river basin; n: Liaohe river basin, Haihe river basin, Yellow river Basin, Songhuajiang river basin, and Huaihe river basin |
| **(Zhang et al., 2022c)** | Yangtze River | 2001-2018 | CCW Model | Climate variability |
| **(Chen et al., 2022)** | Six river basins in China | 1982-2015 | Gray Relational Analysis (GRA) | Precipitation |





| | | | | |
|---|---|---|---|---|
| **(Zhai and Tao, 2021)** | China | 1982-2015 | VIC Model | Climate change |
| **(Li et al., 2020a)** | Yihe River | 1960-2013 | SWAT+WRF | Climate variability |
| **(Shen et al., 2017)** | China | 1960-2010 | Budyko | Underlying surface change (n): the Songhua Basin, the Liaohe Basin and the Haihe Basin; Climate change: in other basins. |

Elasticity analysis (Section 3.4) revealed distinct sensitivities of WY to
environmental drivers: precipitation exhibited the highest elasticity coefficient for the
whole China ($\varepsilon P$ = 1.55), followed by $CO_2$ ($\varepsilon CO_2$ = 0.55) and NDVI ($\varepsilon NDVI$ = -0.44).
However, spatial analysis showed that vegetation and [$CO_2$] collectively dominated
WY changes in 400–1600 mm/yr precipitation zones, despite their lower sensitivity
rankings. This apparent contradiction stemmed from the interplay between elasticity
and the magnitude of driver change. In the 400–1600 mm/yr precipitation zones, NDVI
displayed high spatial heterogeneity (Fig. 8), whereas precipitation fluctuated within a
narrower relative range. Consequently, vegetation's stronger spatiotemporal variability
amplified its hydrological influence, overriding its lower elasticity. Similarly, $CO_2$'s
historical impact was constrained by its slow accumulation rate (0.49%/yr), yet its
relatively high elasticity positions it as a latent driver.

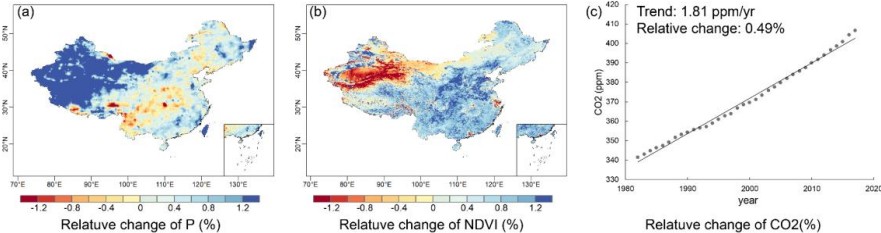

**Figure 8.** Spatial distribution of relative changes of different variables: (a) annual precipitation,
and (b) NDVI.
This historical constraint, however, belied $CO_2$'s transformative potential under
intensified forcing scenarios. CMIP6 SSP585 projections indicate [$CO_2$] will rise at
2.34%/yr—nearly fivefold faster than historical rates (Cheng et al., 2022). At this
trajectory, $CO_2$'s elasticity would drive a +1.29% annual WY increase, eclipsing both
vegetation greening effects and even surpassing precipitation-driven changes in some





regions. Such reversal underscores the imperative to prioritize [$CO_2$] in long-term water
management, particularly in 400–1600 mm/yr precipitation zones.
**4.3   Uncertainty in attribution analysis**
This study provides valuable insights into the relationship between water resources
management and environmental changes, which can guide environmental management
strategies. However, several limitations exist that need to be addressed in future work
to improve the accuracy and robustness of the results.
Firstly, the improved CCW model does not account for the variation and specific
values of $f_i$, assuming $f_i$ is 0. In reality, $f_i$ represents the ratio of interception evaporation
to total evaporation, and in regions with abundant vegetation, $f_i$ is not zero. Despite this,
considering the small change of $f_i$ in the current year (Zhao et al., 2022), its influence
on runoff trends is negligible in our study (Cheng et al., 2017). However, future work
should prioritize its calculation to improve the precision of WY estimates.
Secondly, the complex interrelationships among climate, vegetation, and [$CO_2$]
cannot be fully disentangled. Vegetation exhibits tight biophysical interactions and
feedback with climate, making it difficult to separate the impacts of climate change,
vegetation dynamics, and [$CO_2$] on hydrological responses. Changes in vegetation,
such as NDVI, reflect a combination of climate change, human activities (e.g.,
reforestation and irrigation), and natural vegetation growth. Additionally, vegetation
greening in upwind regions can increase atmospheric moisture, potentially enhancing
precipitation downwind (Zhang et al., 2021a), which may counteract some of the
negative impacts of increased evapotranspiration on local WY. Although the climate
data used in our model may implicitly capture some of these feedbacks, they cannot be
explicitly separated in this analysis. Consequently, our results represent an attempt to
estimate the direct first-order net impacts of climate, vegetation greening, and [$CO_2$]
increase on WY (Zhang et al., 2021b). Future research should adopt more





comprehensive models that consider soil-vegetation-atmosphere interactions to better
differentiate the contributions of each driving factor to WY.
Thirdly, the improved CCW model does not incorporate certain human activities,
such as dam construction and water extraction, which should be incorporated in future
studies. Our research also excludes water bodies and built-up land. While urbanization
can increase flood risks due to the growing proportion of impervious surfaces (Wasko
and Sharma, 2017), these land-use changes represent a small portion of China's land
area.
Finally, the future impact of vegetation greening on hydrological dynamics will
depend on projected climate warming and drying trends, the persistence of vegetation
greening, $[CO_2]$ changes, and the complex feedbacks between climate, soil, and
vegetation. These interactions require long-term study, and future research will involve
more extensive monitoring to better capture these evolving dynamics.

## 561 **5 Conclusions**

In this study, we improved the CCW model incorporating dynamic water use
efficiency (WUE) calculation to explicitly represent $CO_2$-physiological feedback on
water yield. This mechanistic improvement enabled comprehensive national-scale
assessment quantifying the relative contributions of climate forcing, vegetation
structural changes, and $CO_2$-driven stomatal regulation to water yield (WY) dynamics
in China. The main conclusions are as follows:
The improved CCW model effectively simulated WY variations in most basins
under increased $[CO_2]$ scenarios, demonstrating its applicability and reliability in
modeling WY changes.
Climate change, particularly variations in precipitation, emerged as the primary
driver influencing WY, displaying significant regional disparities in its effects.
Vegetation changes constituted the second most critical factor, predominantly resulting



in WY reduction, notably in central China. While the effect of $CO_2$-induced stomatal
closure on WY was comparatively minor. Spatial analysis aligned with isohyetal lines
further revealed that vegetation change and $[CO_2]$ exerted greater influence within the
400–1600 mm precipitation range.
The elasticity analysis of WY indicated that northern basins exhibit higher
sensitivity to influencing factors, whereas southern basins demonstrate relatively lower
elasticity. Specifically, the absolute elasticity coefficients for the whole China were
ranked in descending order as follows: precipitation > $[CO_2]$ > NDVI. Thus,
accelerating $[CO_2]$ rise (2.34% /yr under SSP585) will amplify its hydrological role,
potentially elevating $CO_2$-driven WY increases to +1.29% annually by 2100, surpassing
climate and vegetation impacts.
These insights provide a nuanced understanding of regional hydrological
responses, essential for sustainable water resource management under changing
environmental conditions.
## Acknowledgements
This research was supported by the China National Key R&D Program (grant no.
2024YFF1306901).
## Data Availability Statement
Datasets used for driving models were obtained from different sources described
in Table 1. All the data related to our results in this study can be found online: the NDVI
data (https://doi.org/10.6084/m9.figshare.c.7002225.v1); the climate data
(https://www.tpdc.ac.cn/zh-hans/data/8028b944-daaa-4511-8769-965612652c49/); the
land use and land cover (LULC) data (https://zenodo.org/records/8239305) (Liu et al.,
2023); and the $[CO_2]$ (http://cdiac.esd.ornl.gov/ftp/trends/co2/maunaloa.co2), except
for the streamflow records for hydrological gauging stations, which are available upon
reasonable request.



**Author contributions**

HS designed the study, developed the model code, did the simulation experiments, and wrote the first draft of the paper. HY designed the research and edited the manuscript. CL provided feedback on the results and edited the manuscript.

**Competing interests**

The contact author has declared that neither they nor their co-authors have any competing interests.



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
