# Peer review of "Enhanced understanding of dominant drivers of"

_EGUsphere, 2025_

## Referee Comment (RC1)

This manuscript presents an advancement in ecohydrological modeling by improving a coupled carbon-water model to explicitly incorporate $CO_2$-physiological feedback through water use efficiency. The authors attribute changes in water yield across China to climate, vegetation, and $CO_2$ drivers, and project a dominant role of $CO_2$ under the SSP585 scenario. The study is methodologically sound and addresses a pressing need for better attribution frameworks in hydrological-climate-ecological modeling. However, several conceptual, methodological, and presentation issues must be addressed before the manuscript is suitable for publication.

General Comments

1. Many studies have previously explored the attribution of water yield changes to climate and vegetation drivers. The manuscript should include a concise sentence in the introduction to clearly state how this study *specifically* advances beyond existing work. What is the new insight or capability that prior models or attribution methods could not achieve?

2. The authors should explicitly explain why combining the WUE-related $CO_2$ pathway is necessary in this study. What are the limitations of models that ignore this feedback? Has similar work been done that includes WUE, and if so, how does this model differ? A literature review paragraph in the introduction or methodology section would help justify the novelty and necessity of this approach.

3. The abstract currently reads as a list of findings without a logical flow. It should be restructured to highlight the motivation and gap, the modeling approach (including WUE), the key results (not all numerical), and the main conclusion. The current version is too lengthy and overly descriptive.

4. The manuscript language focuses heavily on reporting numerical results (e.g., spatial patterns, percentages). However, more effort is needed to interpret and discuss the underlying ecohydrological processes, theoretical implications, and model behavior. Avoid a "data-dump" tone; instead, synthesize meaning behind results.

5. The abstract (line 19) states the study analyzes causes of WY changes, but line 74 states the focus is on runoff changes. However, water yield and runoff are not interchangeable. This inconsistency reflects a lack of academic precision and must be corrected throughout the manuscript.

6. Please provide supplementary materials or in-text figures/data validating the accuracy and limitations of the enhanced model that includes WUE. The validation should include metrics like correlation coefficients, RMSE, or NSE for model results with and without WUE. This is essential to assess the credibility of model improvements.

Specific Comments

- Lines 47–49: The first sentence in the introduction requires citation(s) to support the claim being made. Please provide an appropriate reference for the statement.

- Line 115: Please elaborate on how combining the two $CO_2$ response pathways—stomatal conductance and WUE—leads to more accurate conclusions. What are the respective roles of each pathway? What potential biases or benefits arise from

including both in the model compared to only one? The authors are encouraged to provide evidence or theoretical justification here. Since Figure 4a shows model-observation correlation *with* WUE, please also provide a comparison figure for model *without* WUE. This will allow readers to directly assess the added value of including the WUE mechanism.

- Line 238: The claim that "WY is approximately equal to runoff as long-term soil water storage change is negligible" requires citation. This assumption may not hold true in all hydrological settings. Please provide a reference and clearly define both "WY" and "runoff" in the methods section.
- Line 266: Provide a reference for the relative contribution method used. Additionally, clarify how the "trend" term in the equation is calculated.
- Line 276: Justify why a 5% threshold is used to define significance or relevance. Is this based on statistical significance, literature convention, or empirical experience?
- Figure 3a: The colored points in Figure 3a are difficult to read. Please revise the figure format—for example, by increasing symbol size, improving contrast, or separating results by sub-regions or climate zones.
- Lines 458–471: The model description of the stomatal conductance–WUE mechanism is too vague. Please provide explicit equations or cite previous model descriptions to allow reviewers and readers to assess the model's theoretical soundness and parameterization.
- Figure 8: Correct the x-axis label typo: "relatuve" should be "relative".
- Lines 505–513: The discussion on elasticity versus contribution lacks clarity. The authors argue that vegetation and $CO_2$ dominate due to their higher spatial heterogeneity, yet no quantitative evidence is provided to support this claim. Please include relevant statistics comparing the spatial variability of NDVI and precipitation to substantiate the argument.

---

## Author Comment (AC2)

**Response to Reviewer 1 Comments**

Synthesis: This manuscript presents an advancement in ecohydrological modeling by improving a coupled carbon-water model to explicitly incorporate CO2-physiological feedback through water use efficiency. The authors attribute changes in water yield across China to climate, vegetation, and CO2 drivers, and project a dominant role of CO2 under the SSP585 scenario. The study is methodologically sound and addresses a pressing need for better attribution frameworks in hydrological-climate-ecological modeling. However, several conceptual, methodological, and presentation issues must be addressed before the manuscript is suitable for publication.

**Response:** We sincerely appreciate the reviewer's thorough and constructive evaluation of our manuscript. Your summary accurately captured the core objectives and contributions of our work, and we are grateful for the recognition of its methodological advancement and relevance to ecohydrological modeling. We have studied your and the other reviewers' comments carefully and have made corrections/revisions as suggested. The point-to-point responses to the comments and revision are detailed below. In the following, we have detailed how these comments (in black) are raised and our responses (in deep sky blue).

**General comments:**

1: Many studies have previously explored the attribution of water yield changes to climate and vegetation drivers. The manuscript should include a concise sentence in the introduction to clearly state how this study specifically advances beyond existing work. What is the new insight or capability that prior models or attribution methods could not achieve?

Response: Thanks for your constructive comment. To address this point, we revised the introduction to include a single, explicit "what's new" sentence. The revision states that this study advances beyond prior attribution frameworks by embedding [CO2]-dependent, dynamic water-use efficiency (WUE) into the GPP-ET coupling of the CCW model, enabling a mechanistic three-way partition of water-yield/runoff changes into (i) climate change, (ii) vegetation structural change, and (iii) [CO2]-physiological feedback. We clarify the specific pathway—elevated [CO2] lowers stomatal conductance, suppressing transpiration while carbon assimilation (GPP) is not suppressed to the same extent—which raises WUE and prevents the ET reduction from being misattributed to vegetation structure. We also explain how this capability overcomes Budyko-n and regression approaches that conflate vegetation with other catchment properties and fold CO2 effects into PET, thereby delivering physically interpretable attribution that prior methods could not achieve.

Relevant text reads (line 83-91): However, most Budyko-based applications primarily emphasize climate-driven attribution; vegetation and [CO2] influences are typically introduced only indirectly—by assigning temporal changes in "n" to vegetation(Tan et al., 2024; Xue et al., 2022; Zhou et al., 2023) or correlating "n" with NDVI (Liu et al., 2024; Tan et al., 2023), and by embedding [CO2] effects through PET adjustments(Liu et al., 2024). — These practices conflate vegetation with other controls captured by "n" (e.g., soil, topography) and mix [CO2]-physiological impacts with meteorological drivers in PET, making it difficult to isolate

vegetation structural change from [CO2]-induced stomatal adjustments and to ascribe mechanisms robustly (Gan et al., 2021).

(line 119-124): Nevertheless, the original CCW model, while robust in capturing vegetation-climate interactions, adopts a static UWUE and does not account for CO2-induced physiological changes, specifically long-term enhancements in water-use efficiency (WUE) resulting from elevated [CO2], thereby limiting its capacity to isolate [CO2] fertilization effects from vegetation structural and climatic influences (Adams et al., 2020; Li et al., 2023).

(line 125-133): To address this limitation, our study enhanced the CCW framework by incorporating dynamic WUE responses to [CO2], allowing explicit attribution of runoff changes to three distinct drivers: (1) climate change (eg. precipitation, temperature, and so on), (2) vegetation structural change (NDVI, and land use and land cover (LULC)), and (3) [CO2]-physiological feedbacks (stomatal optimization). This extension provides a mechanistically grounded capability that prior empirical or regression-based attribution methods could not achieve, offering new insight into how [CO2] fertilization modulates vegetation–hydrology interactions across large spatial scales.

**A2:** The authors should explicitly explain why combining the WUE-related CO2 pathway is necessary in this study. What are the limitations of models that ignore this feedback? Has similar work been done that includes WUE, and if so, how does this model differ? A literature review paragraph in the introduction or methodology section would help justify the novelty and necessity of this approach.

Response: Thanks for your constructive comment. We revised the introduction and added a targeted literature-review paragraph explaining that frameworks ignoring WUE-related [CO₂] feedbacks systematically (i) misattribute ET change from stomatal closure to vegetation structural change, (ii) conflate [CO₂] physiology with meteorological drivers when routed through PET, and (iii) lose mechanism specificity when vegetation effects are absorbed into a single Budyko-n parameter. We note that prior CCW model typically use static UWUE, which do not capture the [CO₂]-responsive dynamics of the GPP–ET linkage. In contrast, our improved CCW explicitly parameterizes dynamic, [CO₂]-dependent WUE within the GPP–ET coupling, grounded in the stomatal conductance →transpiration suppression/comparatively less-suppressed GPP pathway—thereby delivering physically interpretable, large-scale attribution that separates structural greening from CO₂-physiological effects in a way prior approaches could not.

Relevant text reads (line 92-108): Specifically, elevated  $[CO_2]$  reduces stomatal conductance—due to smaller stomatal apertures and increased leaf resistance (Lammertsma et al., 2011; Xu et al., 2016) which decreases transpiration fluxes (ET). At the same time, carbon assimilation rate (GPP) may increase with higher  $[CO_2]$  availability, but this increase is often less proportional to the reduction in water loss (Montibeller et al., 2022) The resulting imbalance—lower water loss relative to carbon gain—thus leads to higher water-use efficiency (WUE = GPP / ET). In particular, conventional frameworks that neglect  $[CO_2]$ -driven physiological feedbacks fail to represent the enhanced water-use efficiency (WUE) of vegetation under elevated  $[CO_2]$  conditions. This omission leads to ambiguous attribution of runoff variations, as part of the reduction in evapotranspiration induced by stomatal closure is often misinterpreted as a vegetation structural effect rather than a  $[CO_2]$ -induced physiological adjustment. Although

numerous studies have examined vegetation and climate controls on runoff, few have explicitly incorporated the  $[CO_2]$ –WUE feedback within a mechanistic framework. Most existing approaches either completely ignore this feedback or treat it as a simple empirical or linear relationship, rather than capturing its process-based influence on hydrological responses.

**A3:** The abstract currently reads as a list of findings without a logical flow. It should be restructured to highlight the motivation and gap, the modeling approach (including WUE), the key results (not all numerical), and the main conclusion. The current version is too lengthy and overly descriptive.

Response: Thank you for your constructive comment. We have revised the abstract to restore a clear logical flow: it now opens with the motivation and the unresolved gap—disentangling WY contributions from climate, vegetation, and especially  $[CO_2]$ -physiological effects—and then states the modeling approach upfront, specifying the improved CCW framework with dynamic,  $[CO_2]$ -dependent WUE embedded in the GPP–ET coupling to capture the physiological pathway. The key findings are summarized qualitatively rather than as a list of findings. The last sentence states the main conclusion and implication—that dynamic WUE yields a cleaner, mechanistic attribution and decision-relevant insights for water management under future scenarios—while the abstract is substantially condensed, with redundant numerical detail removed to emphasize motivation  $\rightarrow$  method  $\rightarrow$  results  $\rightarrow$  conclusion.

Relevant text reads (line 12-33): The rapid environmental changes, including climate change, escalating atmospheric CO2 concentration ([CO2]), and vegetation dynamics, have been significantly impacting hydrological processes. Yet disentangling the respective contributions of climate, vegetation, and [CO2] change to water yield (WY)-especially clarifying [CO2]driven physiological effects-remains difficult. Therefore, this study improved the coupled carbon and water (CCW) model integrating dynamic water use efficiency (WUE) better capture CO2-physiological feedbacks.; Using scenario analysis, WY changes across China from 1982 to 2017 were attributed to climate, vegetation, and [CO2] drivers. The results showed that climate change (especially precipitation change) emerged as the dominant driver, directly affecting over 70% of China's land area. The vegetation change was the second largest factor to reduce WY, especially in central China. The effect of the escalating [CO2] was relatively small. Spatial analysis aligned with isohyetal lines further revealed that vegetation change and [CO2] exerted greater influence within the 400-1600 mm precipitation zones. In addition, the elasticity analysis showed that the sensitivity ranking of impact factors is precipitation > [CO2] > NDVI for the whole China. Therefore, CMIP6 SSP585 projections indicate that accelerating [CO2] rise will amplify its hydrological effect to a +1.29% annual WY increase by 2100, surpassing vegetation influences. This study refines WY attribution by coupling dynamic WUE with ecohydrological modeling, valuable insights for optimizing regional water resource allocation and developing adaptive ecosystem management strategies under future climate scenarios.

**A4:** The manuscript language focuses heavily on reporting numerical results (e.g., spatial patterns, percentages). However, more effort is needed to interpret and discuss the underlying ecohydrological processes, theoretical implications, and model behavior. Avoid a "data-dump" tone; instead, synthesize meaning behind results.

Response: Thank you for your constructive comment. We revised the Discussion to move from numeric reporting to process-based interpretation and model behavior. The revised discussion explains the ecohydrological mechanisms behind the patterns. At the regional scale (400–1600 mm/yr precipitation zones), we interpret the patterns through coupled water–carbon regulation: vegetation greening elevates transpiration and root water uptake until increasing atmospheric aridity (higher VPD) imposes physiological constraints, while rising [CO2] partially counteracts these constraints by enhancing WUE via stomatal closure, thereby moderating ET and clarifying the observed WY responses. We also link elasticity to the magnitude of driver change to explain net contributions, reconciling why precipitation ranks highest in sensitivity yet vegetation or CO2 can dominate where their relative changes are larger. Model behavior is discussed explicitly—how the improved CCW responds under climate-limited versus water-limited conditions. And We emphasized conceptual synthesis and policy implications (e.g., vegetation management in water-sensitive grids versus climate-adaptation in precipitation-dominated regions).

Relevant text reads (line 465-470): In contrast, our framework mechanistically separates these pathways by explicitly describing the stomatal conductance—WUE relationship based on plant physiological theory. Elevated [CO2] reduces stomatal aperture, thereby lowering stomatal conductance and transpiration flux while only modestly increasing carbon assimilation, leading to an overall enhancement in water-use efficiency (WUE).

(line 485-487): our framework explicitly quantifies CO2's physiological influence on actual evapotranspiration (AET) by mechanistically modeling its role in stomatal conductance and water-use efficiency (WUE).

(line 493-500): This coupled regulation clarified how water and energy jointly constrain evapotranspiration, particularly in 400-1600 mm precipitation zones. In these regions, vegetation growth enhanced transpiration and root water uptake until increasing atmospheric aridity imposed physiological constraints, while rising [CO2] partially counteracted this effect by improving water-use efficiency through stomatal closure.

(line 553-561): From a policy perspective, these spatial contrasts have distinct implications for regional water management. In vegetation-dominated regions such as the Yangtze and Huang river basins, enhancing ecosystem-based restoration, optimizing vegetation composition, and preventing overgreening that may suppress runoff should be prioritized. Conversely, in climate-dominated areas such as Northwest and Southeast China, adaptive measures emphasizing precipitation variability, water storage capacity, and drought resilience are crucial. Recognizing and tailoring water management strategies to these driver-specific regimes can enhance the effectiveness of both ecological restoration and climate adaptation programs across China.

**A5:** The abstract (line 19) states the study analyzes causes of WY changes, but line 74 states the focus is on runoff changes. However, water yield and runoff are not interchangeable. This inconsistency reflects a lack of academic precision and must be corrected throughout the manuscript.

**Response:** Thank you for your constructive comment. We agree that water yield (WY) and runoff are not strictly interchangeable. In this study, WY is defined as precipitation minus evapotranspiration (P–ET), representing the available water output from the land surface at the

annual scale. Under long-term steady-state conditions, when changes in soil water storage are negligible, WY approximates runoff. To ensure conceptual clarity and academic consistency, we have clearly defined WY at its first mention (Section 2.2, Eq. 5) as being approximately equivalent to runoff at the annual scale, and we now use WY consistently in the results and discussion to describe modeled quantities. The term "runoff" is retained only when referring to observed streamflow data used for model validation or to previous studies that explicitly reported results as "runoff". We have revised the abstract to use WY consistently throughout. Relevant text reads (line 22-23): The vegetation change was the second largest factor to reduce WY, especially in central China.

**A6:** Please provide supplementary materials or in-text figures/data validating the accuracy and limitations of the enhanced model that includes WUE. The validation should include metrics like correlation coefficients, RMSE, or NSE for model results with and without WUE. This is essential to assess the credibility of model improvements.

Response: Thank you for your constructive comment. Following the recommendation, we have included a direct validation of the model performance with and without WUE (Fig. 4a and b). While the historical improvement is modest—as expected given that interannual [CO2] variability is small relative to precipitation and that compensating errors can mask process gains—the WUE-enabled formulation corrects a structural deficiency: it captures the physiological pathway whereby elevated [CO2] lowers stomatal conductance and preferentially suppresses transpiration while carbon assimilation is not suppressed to the same extent, thereby preventing ET changes from being misattributed to vegetation structure. This mechanism becomes pivotal under rising-[CO2] scenario, where [CO2] can exceed LAI-driven effects; models without dynamic WUE cannot represent this decoupling of GPP and ET and thus risk biased attributions and projections.

Relevant text reads (line 358-364): As shown in Fig. 4a and b, the observed annual water yield (WY) and the simulated annual WY by the improved CCW model showed strong linear correlations ( $R^2 = 0.7$ ), with the regression line slope being 1.45,  $R^2$  being 0.7, and RMSE being 9.54 mm/year. By contrast, the initial model without WUE showed weaker skill (slope = 1.45,  $R^2 = 0.68$ , RMSE = 9.62 mm·yr-1), indicating that explicitly representing [CO2]-induced regulation of water-use efficiency measurably improves accuracy and reduces bias. Initial picture:

Revised picture:

**Specific Comments**

**A7:** Lines 47–49: The first sentence in the introduction requires citation(s) to support the claim being made. Please provide an appropriate reference for the statement.

**Response:** Thank you for pointing out the problem that the first sentence of the introduction of this draft is not accompanied by a citation. We have added appropriate references at the end of the sentence.

Relevant text reads (line 44-46): The global environment has been undergoing rapid changes, impacting hydrological processes through climate change, escalating atmospheric CO2 concentration [CO2], and vegetation dynamics (Piao et al., 2007; Wei et al., 2024).

A8: Line 115: Please elaborate on how combining the two CO2 response pathways—stomatal conductance and WUE—leads to more accurate conclusions. What are the respective roles of each pathway? What potential biases or benefits arise from including both in the model compared to only one? The authors are encouraged to provide evidence or theoretical justification here. Since Figure 4a shows model observation correlation with WUE, please also provide a comparison figure for model without WUE. This will allow readers to directly assess the added value of including the WUE mechanism.

Response: Thank you for your constructive comment. We realize that our original wording may have caused misunderstanding. The reviewer correctly notes the phrasing about "two pathways" near line 115; however, that description referred to the two vegetation effect pathways in the CCW model, not to two independent CO2 response pathways. In the original Coupled Carbon and Water (CCW) framework, vegetation influences hydrology through a single mechanistic chain in which vegetation structure (NDVI/LAI) controls light absorption and GPP (via FPAR), and evapotranspiration (ET) is coupled to GPP through a biome-specific underlying water-use efficiency (UWUE) term regulated by VPD. These "structural" and "physiological" components describe vegetation-driven effects within the model—not distinct [CO2]-response mechanisms. We have now clarified this explicitly in the revised text. We replaced the phrase "two distinct pathways" with a revised description emphasizing a single vegetation–hydrology coupling chain.

Importantly, the original CCW model did not include an explicit [CO2] pathway. In our improved version, we introduced dynamic WUE responses to [CO2], capturing the physiological feedback whereby elevated [CO2] reduces stomatal conductance, decreases transpiration, and consequently enhances WUE. Thus, there remains only one [CO2] response mechanism—stomatal regulation—whose outcome is expressed as changes in WUE, rather

than two separate [CO2] pathways. We added some sentences explaining the physiological mechanism underlying [CO2]-induced WUE changes.

Regarding the potential biases or benefits of including both effects in the model, we have expanded the Discussion section to more explicitly explain the mechanistic advantages of incorporating dynamic WUE. The revised text clarifies that including [CO2]-induced WUE responses substantially reduces attribution bias and improves physical interpretability of the model. As requested, we now include a direct comparison between simulations with and without the [CO2]-induced WUE adjustment.

Relevant text reads: (line 92-108) Specifically, elevated [CO2] reduces stomatal conductance due to smaller stomatal apertures and increased leaf resistance (Lammertsma et al., 2011; Xu et al., 2016) which decreases transpiration fluxes (ET). At the same time, carbon assimilation rate (GPP) may increase with higher  $[CO_2]$  availability, but this increase is often less proportional to the reduction in water loss (Montibeller et al., 2022) The resulting imbalance—lower water loss relative to carbon gain—thus leads to higher water-use efficiency (WUE = GPP / ET). In particular, conventional frameworks that neglect [CO2]-driven physiological feedbacks fail to represent the enhanced water-use efficiency (WUE) of vegetation under elevated [CO2] conditions. This omission leads to ambiguous attribution of runoff variations, as part of the reduction in evapotranspiration induced by stomatal closure is often misinterpreted as a vegetation structural effect rather than a [CO2]-induced physiological adjustment. Although numerous studies have examined vegetation and climate controls on runoff, few have explicitly incorporated the [CO2]-WUE feedback within a mechanistic framework. Most existing approaches either completely ignore this feedback or treat it as a simple empirical or linear relationship, rather than capturing its process-based influence on hydrological responses. (line 109-124): The coupled carbon and water (CCW) model integrates hydrological and ecological processes by mechanistically linking vegetation dynamics to water and carbon fluxes through remote sensing-driven parameterization (Li et al., 2024b; Zhang et al., 2021b, 2022c). Unlike the Budyko framework's empirical parameter "n" — which conflates vegetation effects with unaccounted catchment characteristics—the CCW model links vegetation and hydrology through a single mechanistic chain. In this framework, vegetation structure (NDVI/LAI) determines canopy absorption of photosynthetically active radiation (FPAR) and hence gross primary production (GPP) via light-use efficiency, while evapotranspiration (ET) is coupled to GPP through a biome-specific underlying water-use efficiency (UWUE) term with vapor pressure deficit (VPD) regulation.. Nevertheless, the original CCW model, while robust in capturing vegetation-climate interactions, adopts a static UWUE and does not account for CO2induced physiological changes, specifically long-term enhancements in water-use efficiency (WUE) resulting from elevated [CO2], thereby limiting its capacity to isolate [CO2] fertilization effects from vegetation structural and climatic influences (Adams et al., 2020; Li et al., 2023). (line 498-500): As a result, the framework provided a more mechanistically grounded understanding of how CO2 fertilization modulates ecosystem water use and hydrological

**A9:** Line 238: The claim that "WY is approximately equal to runoff as long-term soil water storage change is negligible" requires citation. This assumption may not hold true in all

responses at regional scales.

hydrological settings. Please provide a reference and clearly define both "WY" and "runoff" in the methods section.

**Response:** Thank you for your constructive comment. We fully agree with what you pointed out: when "water yield (WY)" and "runoff" are regarded as "WY≈runoff", it is indeed necessary to rigorously explain their applicable conditions and provide references for support.

Relevant text reads (line 236-241): On an annual scale, WY is assumed to be approximately equal to runoff, as changes in soil water storage over long periods (one year or longer) are considered negligible (Xiao et al., 2020; Zhang et al., 2021). Thus, the attribution of WY can also be considered as the attribution of runoff. Accordingly, in this study WY is used as the modelled output, while the term 'runoff' is reserved for observed streamflow or literature values explicitly labelled as such.

**A10:** Line 266: Provide a reference for the relative contribution method used. Additionally, clarify how the "trend" term in the equation is calculated.

**Response:** Thank you for your constructive comment. We have added appropriate references to support the relative contribution method and clarified that the "trend" term in the equation refers to the temporal slope of each variable, calculated using the Theil–Sen method,

**Relevant text reads (line 266-268):** The relative contributions of climate, vegetation, and [CO2] to changes in WY were calculated using the following formula (Ma et al., 2023; Wang et al., 2022):

**A11:** Line 276: Justify why a 5% threshold is used to define significance or relevance. Is this based on statistical significance, literature convention, or empirical experience?

**Response:** Thank you for your constructive comment. The 5% threshold was chosen based on both literature convention and empirical experience. Specifically, Jia et al. (2022) applied a similar 5% floating width criterion when evaluating the equivalence of evapotranspiration products across China, indicating that performance differences within 5% are statistically negligible at the regional scale. Following this convention, we adopted the same tolerance level to distinguish meaningful contributions from minor or uncertain variations.

**Relevant text reads (line 278-280):** If the absolute values of the relative contributions of two factors do not exceed 5%, then these two factors are considered joint significant contributors to the changes in WY at that grid point (Jia et al., 2022).

**A12:** Figure 3a: The colored points in Figure 3a are difficult to read. Please revise the figure format—for example, by increasing symbol size, improving contrast, or separating results by sub-regions or climate zones.

**Response:** Thank you for your constructive comment. In the revised Figure 3a, we have increased the symbol size to enhance readability and visual clarity of the colored points.

Relevant text reads (line 350-351):

Initial picture:

**Revised picture:**

**A13:** Lines 458–471: The model description of the stomatal conductance–WUE mechanism is too vague. Please provide explicit equations or cite previous model descriptions to allow reviewers and readers to assess the model's theoretical soundness and parameterization.

**Response:** Thank you for your constructive comment. We agree that the stomatal conductance—WUE mechanism requires clearer theoretical description. In the revised manuscript, we have added explicit equations and relevant references to clarify the physiological and mathematical basis.

Relevant text reads (line 465-481): In contrast, our framework mechanistically separates these pathways by explicitly describing the stomatal conductance—WUE relationship based on plant physiological theory. Elevated  $[CO_2]$  reduces stomatal aperture, thereby lowering stomatal conductance and transpiration flux while only modestly increasing carbon assimilation, leading to an overall enhancement in water-use efficiency (WUE). This process is represented by the Medlyn-type stomatal conductance model (Medlyn et al., 2011), which links photosynthetic rate (A), transpiration (T), and vapor pressure deficit (D) as:

$$\frac{A}{T} = \frac{C_a P_a}{1.6 \left(D + g_1 \sqrt{D}\right)}$$

where  $C_a$  is atmospheric CO2 concentration,  $P_a$  is air pressure, D is vapor pressure deficit, and  $g_1$  is an empirical slope parameter that quantifies plant sensitivity to CO2 and humidity. According to this formulation, rising [CO2] increases while reducing stomatal conductance, which in turn suppresses transpiration more strongly than photosynthesis, resulting in higher

WUE. This mechanistic representation enables our framework to capture the direct physiological CO2 effect on evapotranspiration, which is otherwise masked in Budyko-type models where CO2 impacts are embedded implicitly in PET or the "n" parameter.

**A14:** Figure 8: Correct the x-axis label typo: "relatuve" should be "relative".

**Response:** Thank you for your constructive comment. The typo in the x-axis label of Figure 8 has been corrected from "relative" to "relative" in the revised version.

Relevant text reads (line 542-543):

**Initial picture:**

**Revised picture:**

**A15:** Lines 505–513: The discussion on elasticity versus contribution lacks clarity. The authors argue that vegetation and CO2 dominate due to their higher spatial heterogeneity, yet no quantitative evidence is provided to support this claim. Please include relevant statistics comparing the spatial variability of NDVI and precipitation to substantiate the argument.

**Response:** We acknowledge that our previous wording "higher spatial heterogeneity" was misleading. Our intention was not to refer to spatial variability but to the relative magnitude of temporal change in vegetation (NDVI) and precipitation within the 400–1600 mm/yr precipitation zones. We have revised the text to clarify this point and to better explain the interplay between elasticity and the relative magnitude of driver change.

**Relevant text reads (line 536-540):** In the 400–1600 mm/yr precipitation zones, NDVI displayed (Fig. 8) a larger relative temporal variation compared with precipitation, which fluctuated within a narrower range. Consequently, vegetation's stronger relative change amplified its hydrological influence, overriding its lower elasticity.

**Reference**

Adams, M. A., Buckley, T. N., and Turnbull, T. L.: Diminishing CO2-driven gains in water-use efficiency of global forests, Nat. Clim. Chang., 10, 466–471, https://doi.org/10.1038/s41558-020-0747-7, 2020.

- Jia, Y., Li, C., Yang, H., Yang, W., and Liu, Z.: Assessments of three evapotranspiration products over China using extended triple collocation and water balance methods, Journal of Hydrology, 614, 128594, https://doi.org/10.1016/j.jhydrol.2022.128594, 2022.
- Lammertsma, E. I., Boer, H. J. D., Dekker, S. C., Dilcher, D. L., Lotter, A. F., and Wagner-Cremer, F.: Global CO2 rise leads to reduced maximum stomatal conductance in Florida vegetation, Proc. Natl. Acad. Sci. U.S.A., 108, 4035–4040, https://doi.org/10.1073/pnas.1100371108, 2011.
- Li, F., Xiao, J., Chen, J., Ballantyne, A., Jin, K., Li, B., Abraha, M., and John, R.: Global water use efficiency saturation due to increased vapor pressure deficit, Science, 381, 672–677, https://doi.org/10.1126/science.adf5041, 2023.
- Liu, C., Feng, S., Zhang, Q., Hu, J., Ma, N., Ci, H., Kong, D., and Gu, X.: Critical influence of vegetation response to rising CO2 on runoff changes, Science of The Total Environment, 906, 167717, https://doi.org/10.1016/j.scitotenv.2023.167717, 2024.
- Ma, T., Wang, T., Yang, D., and Yang, S.: Impacts of vegetation restoration on water resources and carbon sequestration in the mountainous area of Haihe River basin, China, Science of The Total Environment, 869, 161724, https://doi.org/10.1016/j.scitotenv.2023.161724, 2023.
- Medlyn, B. E., Duursma, R. A., Eamus, D., Ellsworth, D. S., Prentice, I. C., Barton, C. V. M., Crous, K. Y., De Angelis, P., Freeman, M., and Wingate, L.: Reconciling the optimal and empirical approaches to modelling stomatal conductance: RECONCILING OPTIMAL AND EMPIRICAL STOMATAL MODELS, Global Change Biology, 17, 2134–2144, https://doi.org/10.1111/j.1365-2486.2010.02375.x, 2011.
- Montibeller, B., Marshall, M., Mander, Ü., and Uuemaa, E.: Increased carbon assimilation and efficient water usage may not compensate for carbon loss in European forests, Commun Earth Environ, 3, 194, https://doi.org/10.1038/s43247-022-00535-1, 2022.
- Piao, S., Friedlingstein, P., Ciais, P., De Noblet-Ducoudré, N., Labat, D., and Zaehle, S.: Changes in climate and land use have a larger direct impact than rising CO2 on global river runoff trends, Proc. Natl. Acad. Sci. U.S.A., 104, 15242–15247, https://doi.org/10.1073/pnas.0707213104, 2007.
- Wang, D. L., Feng, H. M., Zhang, B. Z., Wei, Z., and Tian, Y. L.: Quantifying the impacts of climate change and vegetation change on decreased runoff in china's yellow river basin, Ecohydrology & Hydrobiology, 22, 310–322, https://doi.org/10.1016/j.ecohyd.2021.10.002, 2022.
- Wei, H., Zhang, Y., Huang, Q., Chiew, F. H. S., Luan, J., Xia, J., and Liu, C.: Direct vegetation response to recent CO2 rise shows limited effect on global streamflow, Nat Commun, 15, 9423, https://doi.org/10.1038/s41467-024-53879-x, 2024.
- Xiao, M., Gao, M., Vogel, R. M., and Lettenmaier, D. P.: Runoff and Evapotranspiration Elasticities in the Western United States: Are They Consistent With Dooge's Complementary Relationship?, Water Resources Research, 56, e2019WR026719, https://doi.org/10.1029/2019WR026719, 2020.
- Xu, Z., Jiang, Y., Jia, B., and Zhou, G.: Elevated-CO2 Response of Stomata and Its Dependence on Environmental Factors, Front. Plant Sci., 7, https://doi.org/10.3389/fpls.2016.00657, 2016.
- Zhang, J., Zhang, Y., Sun, G., Song, C., Dannenberg, M. P., Li, J., Liu, N., Zhang, K., Zhang, Q., and Hao, L.: Vegetation greening weakened the capacity of water supply to China's South-to-North Water Diversion Project, Hydrol. Earth Syst. Sci., 25, 5623–5640, https://doi.org/10.5194/hess-25-5623-2021, 2021.

---

## Author Comment (AC3)

**Response to Reviewer 2 Comments**

Synthesis: This manuscript presents an improved Coupled Carbon and Water (CCW) model incorporating dynamic water use efficiency (WUE) to disentangle the effects of climate change, vegetation dynamics, and atmospheric CO2 on water yield (WY) across China during 1982–2017. The study addresses an important research gap by explicitly accounting for CO2 effects and providing a robust attribution analysis at both national and regional scales. The integration of scenario-based attribution and elasticity analysis is innovative and valuable for water resource management and climate adaptation strategies.

The study is generally well-structured, with clear objectives, methods, and results. However, there are a few areas where further clarity, elaboration, and enhancement could improve the overall impact and rigor of the paper. I provide specific comments and suggestions below:

Response: Thank you very much for your thorough and constructive evaluation of our manuscript. We sincerely appreciate the time and effort you have devoted to assessing our work. Your positive recognition of the study's innovation—particularly the improved Coupled Carbon and Water (CCW) model with dynamic water use efficiency (WUE) and the integration of scenario-based attribution with elasticity analysis—encourages us greatly. We have carefully considered all your valuable comments and those from the other reviewers, and we have revised the manuscript accordingly to enhance its clarity, depth, and scientific rigor. A detailed, point-by-point response to all comments is provided below, where the reviewers' comments are presented in black and our responses are provided in deep sky blue.

**Abstract:**

1: The abstract effectively summarizes the study, but it is rather dense with technical terms and numerical results. Consider slightly rebalancing it by adding one or two sentences that emphasize the practical implications (e.g., relevance for water resource management and ecological restoration) so that non-specialist readers can more easily grasp the significance.)

**Response:** Thank you for your constructive comment. We agree that the abstract was originally dense with technical terms and numerical results, which may limit accessibility for non-specialist readers. To improve readability and highlight the broader significance, we have added one sentence at the end of the abstract emphasizing the practical implications of our findings for water resource management and ecological restoration. The revised abstract now provides a more balanced presentation between technical content and applied relevance.

Relevant text reads (line 12-33): The rapid environmental changes, including climate change, escalating atmospheric CO2 concentration ([CO2]), and vegetation dynamics, have been significantly impacting hydrological processes. Yet disentangling the respective contributions of climate, vegetation, and [CO2] change to water yield (WY)—especially clarifying [CO2]-driven physiological effects—remains difficult. Therefore, this study improved the coupled carbon and water (CCW) model integrating dynamic water use efficiency (WUE) better capture CO2-physiological feedbacks.; Using scenario analysis, WY changes across China from 1982 to 2017 were attributed to climate, vegetation, and [CO2] drivers. The results showed that climate change (especially precipitation change) emerged as the dominant driver, directly affecting

over 70% of China's land area. The vegetation change was the second largest factor to reduce WY, especially in central China. The effect of the escalating [CO2] was relatively small. Spatial analysis aligned with isohyetal lines further revealed that vegetation change and [CO2] exerted greater influence within the 400–1600 mm precipitation range. In addition, the elasticity analysis showed that the sensitivity ranking of impact factors is precipitation > [CO2] > NDVI for the whole China. Therefore, CMIP6 SSP585 projections indicate that accelerating [CO2] rise will amplify its hydrological effect to a +1.29% annual WY increase by 2100, surpassing vegetation influences. This study refines WY attribution by coupling dynamic WUE with ecohydrological modeling, valuable insights for optimizing regional water resource allocation and developing adaptive ecosystem management strategies under future climate scenarios.

**Introduction and Background:**

1: While the introduction clearly outlines the motivation, it would be useful to more explicitly highlight what distinguishes this study from other model applications. For example, a brief comparison with previous studies could emphasize the novelty of incorporating CO2-induced WUE changes.

Response: Thank you for your constructive comment. We agree that the introduction should more explicitly highlight what distinguishes this study from existing model applications. In the revised version, we have clearly articulated that the novelty of our study lies in the mechanistic incorporation of CO2-driven dynamic water-use efficiency (WUE) feedback into the coupled carbon and water (CCW) model.

Relevant text reads (line 119-133): Nevertheless, the original CCW model, while robust in capturing vegetation-climate interactions, does not account for CO2-induced physiological changes, specifically long-term enhancements in water-use efficiency (WUE) resulting from elevated [CO2], thereby limiting its capacity to isolate [CO2] fertilization effects from vegetation structural and climatic influences(Adams et al., 2020; Li et al., 2023).

To address this limitation, our study enhanced the CCW framework by incorporating dynamic WUE responses to [CO2], allowing explicit attribution of runoff changes to three distinct drivers—climate change (eg. precipitation, temperature, and so on), vegetation structural change (NDVI, and land use and land cover (LULC)), and [CO2]-physiological effects (stomatal optimization). This extension provides a mechanistically grounded capability that prior empirical or regression-based attribution methods could not achieve, offering new insight into how [CO2] modulates vegetation—hydrology interactions across large spatial scales.

2: The introduction mainly focuses on China, but since similar issues of climate–vegetation–CO2 interactions exist globally, it may help to briefly situate this work in a broader international context. For example, mentioning comparable studies in other semi-arid or monsoon-influenced regions would show the wider relevance of the improved CCW model.

Response: Thank you for your constructive comment. We agree that positioning the study within a broader international context would strengthen the relevance and generalizability of our work. In the revised Introduction, we have added a concise paragraph highlighting that similar interactions among climate, vegetation, and CO2 have been widely reported across other semi-arid and monsoon-influenced regions worldwide—such as the Sahel, South Asia, and Mediterranean ecosystems—where vegetation greening and water yield responses to climate and CO2 forcing have been actively studied. This addition clarifies that, although our

analysis focuses on China, the improved CCW model and its explicit integration of CO2-induced WUE feedbacks are broadly applicable to global ecohydrological research and water resource management in similar climatic zones.

Relevant text reads (line 44-61): The global environment has been undergoing rapid changes, impacting hydrological processes through climate change, escalating atmospheric CO2 concentration [CO2], and vegetation dynamics (Piao et al., 2007; Wei et al., 2024). Notably, China has experienced a visible greening trend in recent decades, prompting a heightened focus on ecological and water resource concerns (Chen et al., 2019). Investigating the influence of vegetation changes on runoff has thus emerged as a pivotal research area, aligning with China's increasing emphasis on environmental sustainability. China's diverse climatic zones and pronounced greening make it an ideal natural laboratory for investigating these ecohydrological feedbacks, with insights that are globally relevant yet directly informative for sustainable water resource management and ecological restoration in China(Ogutu et al., 2021; Yang et al., 2019), and for other semi-arid and monsoon-influenced regions such as the Sahel, South Asia, and the Mediterranean Basin(Nkiaka et al., 2025; Rahman et al., 2025; Serrano-Notivoli et al., 2022). Understanding the intricate interplay among vegetation dynamics, climate change, and [CO2] within the water cycle, particularly concerning runoff therefore it is not only of global relevance but also of profound importance for advancing sustainable water resource management and ecological restoration strategies in China under accelerating environmental change.

**Methods:**

1: The improved CCW model assumes interception evaporation factor fi equals zero. Since this simplification is acknowledged in the Discussion, please provide a short justification earlier in the Methods section so that readers can immediately understand this limitation.

**Response:** Thank you for your constructive comment. We agree that the assumption of the interception evaporation factor  $(f_i)$  being set to zero should be briefly justified in the Methods section to enhance transparency and reader understanding. We have accordingly revised the section to clarify this simplification and its rationale.

Relevant text reads (line 218-224): In this study, the interception evaporation factor (fi) was set to zero. This simplification follows previous large-scale ecohydrological studies (Cheng et al., 2017), which reported that canopy interception and soil surface evaporation account for a minor portion of total evapotranspiration at annual to multi-decadal scales. Given that the improved CCW model focused on yearly water yield (WY) dynamics rather than event-scale hydrological responses, neglecting interception loss reduces model complexity without substantially affecting WY estimation.

2: The attribution analysis is based on "trends" in WY under different scenarios, but the exact method of calculating these trends (e.g., linear regression, Mann–Kendall test, or another approach) is not clearly described. Providing details on the trend detection method, as well as the statistical significance criteria, would help readers better assess the robustness of the results. **Response:** Thank you for your constructive comment. We agree that the description of trend estimation could be made clearer. Specifically, we quantified the long-term trend in annual WY (1982–2017) using the non-parametric Theil–Sen estimator for the slope.

**Relevant text reads (line 267-270):** For each scenario, the long-term trend in annual WY over 1982–2017 was quantified using the Theil–Sen estimator, yielding a robust slope. The relative contributions of climate, vegetation, and [CO2] to changes in WY were calculated using the following formula:

**Results:**

1: Figures 5 and 6 demonstrate spatial heterogeneity in WY drivers. It would help if the authors could provide a more policy-relevant interpretation, e.g., what the findings imply for water resource planning in regions where vegetation dominates versus where climate dominates.

**Response:** Thank you for your constructive comment. We agree that linking spatial heterogeneity in WY drivers to practical implications can improve the relevance of our findings. Since the Results section primarily presents objective spatial patterns, we have added a short paragraph in the Discussion section to interpret the regional contrasts from a management perspective.

Relevant text reads (line 555-563): From a policy perspective, these spatial contrasts have distinct implications for regional water management. In vegetation-dominated regions such as the Yangtze and Huang river basins, enhancing ecosystem-based restoration, optimizing vegetation composition, and preventing overgreening that may suppress runoff should be prioritized. Conversely, in climate-dominated areas such as Northwest and Southeast China, adaptive measures emphasizing precipitation variability, water storage capacity, and drought resilience are crucial. Recognizing and tailoring water management strategies to these driver-specific regimes can enhance the effectiveness of both ecological restoration and climate adaptation programs across China.

2: The elasticity analysis provides valuable insights into the sensitivity of WY to different drivers. However, the discussion could be enhanced by more explicitly linking the elasticity results to the relative contributions of each driver. For instance, why does CO2 have a higher elasticity than NDVI yet a smaller overall contribution? Clarifying how elasticity and the magnitude of change jointly determine the net impact would strengthen the interpretation.

**Response:** Thank you for your constructive comment. We agree that the linkage between elasticity and contribution deserves clarification. In the revised manuscript, we emphasize that elasticity quantifies the sensitivity of WY to a unit change in each driver, whereas contribution reflects the integrated effect of both elasticity and the magnitude of driver change.

Relevant text reads (line 532-543): Elasticity analysis (Section 3.4) revealed distinct sensitivities of WY to environmental drivers: precipitation exhibited the highest elasticity coefficient for the whole China ( $\epsilon P = 1.55$ ), followed by CO2 ( $\epsilon CO_2 = 0.55$ ) and NDVI ( $\epsilon NDVI = -0.44$ ). However, spatial analysis showed that vegetation and [CO2] collectively dominated WY changes in 400–1600 mm/yr precipitation zones, despite their lower sensitivity rankings. The joint effect of elasticity and the magnitude of driver change that determines each driver's net contribution. In the 400–1600 mm/yr precipitation zones, NDVI displayed (Fig. 8) a larger relative temporal variation compared with precipitation, which fluctuated within a narrower range.. Consequently, vegetation's stronger relative change amplified its hydrological influence, overriding its lower elasticity. Similarly, CO2's historical impact was constrained by its slow accumulation rate (0.49%/yr), yet its relatively high elasticity positions it as a latent driver.

**Discussion:**

1: While the study focuses on climate, vegetation, and CO2 drivers, other human activities such as reservoir regulation, irrigation, and groundwater extraction can also significantly affect water yield in China. Since these processes are briefly mentioned as limitations, it would strengthen the discussion if the authors could add a short paragraph acknowledging how such anthropogenic factors may interact with the modeled drivers, and whether the improved CCW framework could potentially incorporate them in future work.

**Response:** Thank you for your constructive comment. We agree that anthropogenic factors such as reservoir regulation, irrigation, and groundwater extraction play an important role in shaping hydrological responses in China. To address this point, we have expanded the existing discussion paragraph.

Relevant text reads (line 590-599): Thirdly, the improved CCW model does not incorporate certain human activities, such as large-scale irrigation, groundwater pumping, and reservoir regulation, which should be incorporated in future studies. For instance, irrigation can sustain vegetation greening during dry seasons, potentially amplifying the vegetation–climate feedback on water yield. Incorporating such anthropogenic processes into the CCW framework through coupled irrigation and water management modules would enable more comprehensive attribution analyses in future studies.

**Reference**

- Adams, M. A., Buckley, T. N., and Turnbull, T. L.: Diminishing CO2-driven gains in water-use efficiency of global forests, Nat. Clim. Chang., 10, 466–471, https://doi.org/10.1038/s41558-020-0747-7, 2020.
- Chen, C., Park, T., Wang, X., Piao, S., Xu, B., Chaturvedi, R. K., Fuchs, R., Brovkin, V., Ciais, P., Fensholt, R., Tømmervik, H., Bala, G., Zhu, Z., Nemani, R. R., and Myneni, R. B.: China and India lead in greening of the world through land-use management, Nat Sustain, 2, 122–129, https://doi.org/10.1038/s41893-019-0220-7, 2019.
- Cheng, L., Zhang, L., Wang, Y.-P., Canadell, J. G., Chiew, F. H. S., Beringer, J., Li, L., Miralles, D. G., Piao, S., and Zhang, Y.: Recent increases in terrestrial carbon uptake at little cost to the water cycle, Nat Commun, 8, 110, https://doi.org/10.1038/s41467-017-00114-5, 2017.
- Li, F., Xiao, J., Chen, J., Ballantyne, A., Jin, K., Li, B., Abraha, M., and John, R.: Global water use efficiency saturation due to increased vapor pressure deficit, Science, 381, 672–677, https://doi.org/10.1126/science.adf5041, 2023.
- Nkiaka, E., Bryant, R. G., and Dembélé, M.: Quantifying Sahel Runoff Sensitivity to Climate Variability, Soil Moisture and Vegetation Changes Using Analytical Methods, Earth Syst Environ, 9, 491–504, https://doi.org/10.1007/s41748-024-00464-3, 2025.
- Ogutu, B. O., D'Adamo, F., and Dash, J.: Impact of vegetation greening on carbon and water cycle in the African Sahel-Sudano-Guinean region, Global and Planetary Change, 202, 103524, https://doi.org/10.1016/j.gloplacha.2021.103524, 2021.
- Rahman, G., Farooq, U., Jung, M.-K., and Kwon, H.-H.: Spatiotemporal vegetation dynamics in South Asia (2001-2023): roles of climate and anthropogenic activities, Geosci. Lett., 12, 31, https://doi.org/10.1186/s40562-025-00403-8, 2025.
- Serrano-Notivoli, R., Martínez-Salvador, A., García-Lorenzo, R., Espín-Sánchez, D., and Conesa-García, C.: Rainfall–runoff relationships at event scale in western Mediterranean ephemeral streams, Hydrol. Earth Syst. Sci., 26, 1243–1260, https://doi.org/10.5194/hess-26-1243-2022, 2022.
- Yang, Y., Roderick, M. L., Zhang, S., McVicar, T. R., and Donohue, R. J.: Hydrologic implications of vegetation response to elevated CO2 in climate projections, Nature Clim Change, 9, 44–48, https://doi.org/10.1038/s41558-018-0361-0, 2019.